# Fibrotic microtissue array to predict anti-fibrosis drug efficacy

Mohammadnabi Asmani [1], Sanjana Velumani [1], Yan Li[1], Nicole Wawrzyniak [1], Isaac Hsia [1], Zhaowei Chen[1], Boris Hinz [2,3] & Ruogang Zhao [1]

Fibrosis is a severe health problem characterized by progressive stiffening of tissues which causes organ malfunction and failure. A major bottleneck in developing new anti-fibrosis therapies is the lack of in vitro models that recapitulate dynamic changes in tissue mechanics during fibrogenesis. Here we create membranous human lung microtissues to model key biomechanical events occurred during lung fibrogenesis including progressive stiffening and contraction of alveolar tissue, decline in alveolar tissue compliance and traction force-induced bronchial dilation. With these capabilities, we provide proof of principle for using this fibrotic tissue array for multi-parameter, phenotypic analysis of the therapeutic efficacy of two anti-fibrosis drugs recently approved by the FDA. Preventative treatments with Pirfenidone and Nintedanib reduce tissue contractility and prevent tissue stiffening and decline in tissue compliance. In a therapeutic treatment regimen, both drugs restore tissue compliance. These results highlight the pathophysiologically relevant modeling capability of our novel fibrotic microtissue system.

[1] Department of Biomedical Engineering, University at Buffalo, The State University of New York, Buffalo, NY 14260, USA. [2] Laboratory of Tissue Repair and Regeneration, Matrix Dynamics Group, Faculty of Dentistry, University of Toronto, Toronto, ON M5S 3E2, Canada. [3] Institute of Biomaterials & Biomedical Engineering, University of Toronto, Toronto, ON M5S 3G9, Canada. Correspondence and requests for materials should be addressed to R.Z. (email: rgzhao@buffalo.edu)

Lung fibrosis, such as idiopathic pulmonary fibrosis (IPF) or radiation-induced pulmonary fibrosis, frequently leads to organ failure and death and currently has no no cure[1, 2]. Candidates for anti-fibrotic therapies such as inhibitors directed against TGF-β and LoxL2 have been identified; however, translation of these laboratory discoveries into clinical practice is facing major obstacles[1,3]. The main bottlenecks are slow progression of the disease, high cost and the need for large sample sizes associated with animal studies and clinical trials. To justify the development of new therapies in prolonged and expensive clinical trials, in vitro screening assays are needed to provide early evidence of the efficacy of anti-fibrotic therapies.

Progression of lung fibrosis is characterized by stiffening of the membranous tissue that makes up the alveolar air sacs, caused by the accumulation of highly contractile and collagen-producing myofibroblasts[4–6]. To model the physio-pathological characteristics of lung fibrosis in vitro, different experimental systems have been developed. Planar 2D culture-based assays such as high-content screening performed in multi-well plates offer image-based, multi-parameter readout of fibrotic cellular markers, but they have very limited capacity to measure or reproduce tissue mechanics[7–9]. Centimeter-scale, thick-engineered tissues are not suitable for the drug screening due to inherent low-throughput and diffusional limitations[10–12]. Recently, microfabricated culture systems such as organ-on-chip systems have emerged as high-throughput and resource-saving approaches for drug screening[13–15]. These systems, to some extent, mimic the structure and composition of human tissues and allow to test drug responses under in vivo-like conditions. In an existing lung-on-chip device, an alveolar epithelial cell-lined air chamber and a capillary endothelial cell-lined fluidic chamber is separated by a porous silicone membrane to mimic the alveolar–capillary interface[13]. This system is able to model inflammatory responses and pulmonary edema and has been used to identify potential therapeutics for edema[16]. However, extracellular matrix (ECM) components are lacking in this construct which eliminates the tissue remodeling aspect of fibrosis. This limits the capability of this device to modulate and/or assess tissue mechanical properties and reduces the utility in pulmonary fibrosis studies.

To address these limitations that are common to all planar cell cultures adhering to 2D substrates, we and others have developed arrays of 3D microtissues that are suspended over multiple flexible poly(dimethylsiloxane) (PDMS) micropillars[17–19]. These microtissues form through cell-mediated ECM remodeling and undergo morphological and structural changes that are guided by micropillar-defined mechanical boundary conditions. The system's ability to control tissue morphogenesis opens up possibilities to engineer biomimetic tissue morphology and structures that are important to tissue's biological functions. Previous designs focused on rod-shaped microtissues where uniaxial stress was achieved to facilitate the alignment of cells along tissue's longitudinal axis. Such stress-mediated cell alignment has been shown to promote the development and maturation of engineered skeletal and cardiac muscles[20–22]. In another study, thick microtissue mimicking the stromal tissue layer of the skin were created to study the fibroblast migration during the closure of an artificial wound opening[20]. However, these microtissue designs do not recapitulate the thin, membranous morphology of lung alveolar tissue. This specific architecture is important for the development of pathological features in fibrotic lung, such as the decline in tissue compliance and traction force-induced bronchial dilation[5,23].

By controlling the morphogenic process of suspended microtissues, here we create membranous lung microtissues that recapitulate key biomechanical properties of both healthy and fibrotic lung alveolar tissues. The microtissue is composed of lung fibroblast-populated collagen matrix that allows robust response to fibrosis induction and therapy. We demonstrate pathological transition of this originally compliant, membranous lung interstitial microtissue to fibrotic stiff tissue, caused by contraction and activation of myofibroblasts. Further integration of this fibrotic microtissue with patterned stress distribution and tissue stretching allow the modeling of other pathological features of the fibrotic lung, such as the decline in tissue compliance and traction force-induced bronchial dilation. In proof of principle tests, we then document the utility of this system for multi-parameter, phenotypic analysis of the therapeutic efficacy of two FDA-approved anti-fibrosis drugs, Pirfenidone and Nintedanib. While Pirfenidone and Nintedanib have different inhibitory effects on myofibroblast activation and tissue contractility in preventative treatments, both drugs effectively inhibit fibrotic tissue stiffening and decline in tissue compliance. Both anti-fibrosis drugs restore tissue compliance to different degree in a therapeutic treatment regimen. Together, these studies highlight the pathophysiologically relevant modeling capability of our innovative fibrotic microtissue system. The application of this system will not only expedite the efficacy analysis of anti-fibrotic therapies but also help to unveil their potential mode of action.

## Results

**Development of arrays of membranous lung microtissues.** Human lung alveoli are interlinked air sacs, each surrounded by membranous interstitial tissue with large surface area and small thickness. This unique morphology, characterized by high span length ($S$) to thickness ($t$) ratio that ranges from 30:1 to 50:1 (Fig. 1a)[24,25], not only facilitates gas exchange but is also critical to the development of pathological features during lung fibrosis. To recapitulate this unique alveolar morphology, we created membranous human lung microtissues by leveraging the dynamic remodeling capability of a novel microtissue system (Fig. 1b). Using finite element (FE) models, we first determined how microtissue design, geometry, and boundary conditions will affect the formation of membranous yet structurally stable microtissues. We explored various microtissue designs, including a large-size, multi-leaflet design that mimics the shape of multiple alveoli and single-leaflet designs with varied sizes (Fig. 1c; Supplementary Fig. 1). FE models of microtissues were constructed using a constitutive formulation that allows for large tissue deformation under cell-generated isotropic contractile force[26]. We showed that under such contractile forces, tissue compacted freely through its thickness (Z-axis, Supplementary Movie 1), but the compaction in X–Y plane was restricted by micropillar-defined boundary conditions, leading to stress concentration, a localized increase in stress, around the micropillars (Supplementary Movie 2). We observed that these high stresses dissipated towards the tissue center along the diagonal axis of the microtissue, as shown by the direction of the principal stress vectors in highlighted area (Fig. 1c). Comparison between different microtissue designs showed that although large-size and medium-size designs both result in relatively high $S/t$ ratios, which are ideal to model alveolar tissues, they are associated with different levels of stress concentrations that may affect the structural integrity of the microtissues differently (Fig. 1c; Supplementary Figs. 1, 2). Therefore, further experimental validation of these microtissue designs is needed to determine their actual performance.

To experimentally validate the microtissue designs, we fabricated micropillar arrays using multi-layer photolithography method and following the geometries defined in the large-size and medium-size FE models (Supplementary Fig. 3). When mixtures of human lung small airway epithelial cells (SAECs) or lung fibroblasts with ECM proteins were placed in individual

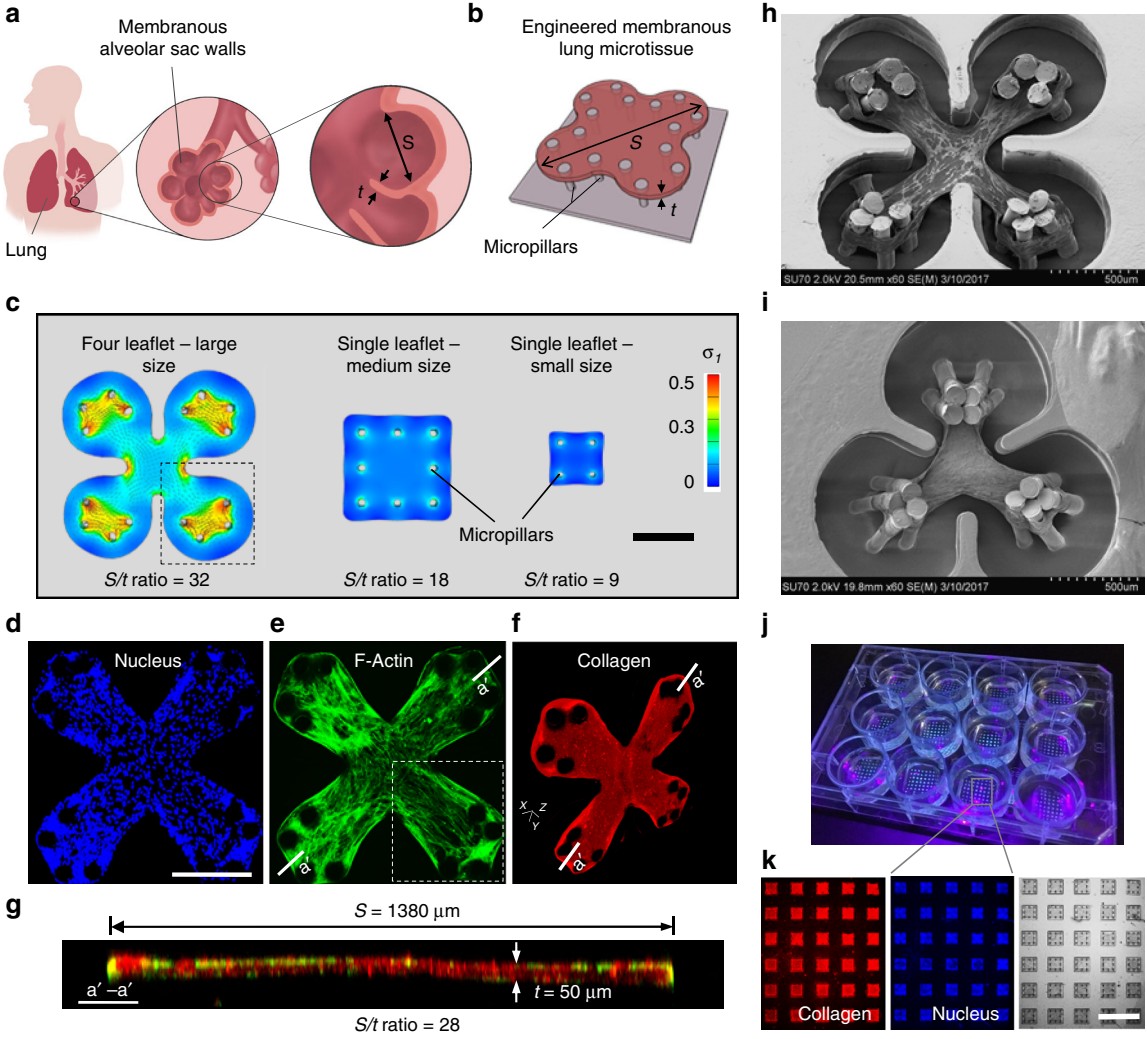

**Fig. 1** Development of arrays of membranous lung microtissues. **a** The lung alveolar sac wall features large surface area and small tissue thickness, which is characterized by the high span length (*S*) to thickness (*t*) ratio. **b** Engineered lung microtissues were developed to model the membranous morphology of the alveolar sac wall. **c** Finite element (FE) models were developed to study the effects of microtissue geometry and size on the evolution of contractile stress during microtissue formation. First principal stress contour was plotted on deformed microtissue geometry with overlaid plot of stress vectors, where applicable. Scale bar is 500 μm. Representative 2D-projected fluorescent confocal images of an experimentally-created four-leaflet lung microtissue stained for nucleus (**d**) and F-actin (**e**). Note highlighted area shows highly aligned F-actin stress fibers running along the diagonal axis of the microtissue, matching with the direction of the principal stress in highlighted area in **c**. Scale bar in **d** is 500 μm. **f** 3D isometric view of the same microtissue stained for collagen type-I. **g** Merged fluorescent cross-sectional view (F-actin and collagen type-I) taken at *a′*–*a′* plane of the four-leaflet microtissue showing *S/t* ratio of 28, corresponding well to alveolar sac geometry. SEM images of a four leaflet (**h**) and a three leaflet (**i**) human lung fibroblast-populated microtissue. **j** Arrays of square lung microtissues were integrated into a 12-well plate to enable parallel testing of multiple pharmacological conditions. Bright spots in each well correspond to individual microtissues. **k** Immunofluorescence imaging of the microtissue array allowed multi-parameter, phenotypic analysis of the drug efficacy. Scale bar is 3 mm

microwells, cells started to spread and compact the ECM (Supplementary Movie 3), leading to the formation of shape-defined, membranous microtissues (Fig. 1d–f, Supplementary Fig. 4). We observed that actin stress fibers of cells within the leaflets of a multi-leaflet microtissue aligned along the diagonal axis of the microtissue, matching with the direction of the principal stress. This suggests that actomyosin-based cellular contractility drove tissue morphogenesis under pre-defined boundary conditions, as predicted by the FE model. Cross-sectional measurement of the microtissue demonstrates a thickness ranging from 35 to 55 μm. For a large-size, four-leaflet microtissue with a diagonal span length of 1380 μm, the resulting *S/t* ratio is 28:1, corresponding well to alveolar sac geometry (Fig. 1g). In addition to the four-leaflet design (Fig. 1h),

we also tested large three-leaflet design (Fig. 1i; Supplementary Fig. 4) and medium-size single-leaflet designs. While these designs all supported the formation of microtissues, the rate of microtissue failure, as measured by the rates of micropillar breakage and microtissue detachment from the micropillar head, increased with increased design complexity and microtissue size. Therefore, we chose the medium-size, single-leaflet design for our drug screening system to ensure robust performance and easy handling. To enable parallel testing of multiple pharmacological conditions, we created microtissue arrays in a 12-well plate format with 8 × 8 microtissues per well and a total of 768 microtissues per plate (Fig. 1j, k). The large number of samples enabled multi-parameter, phenotypic analysis of the drug candidates with a throughput higher than conventional assays.

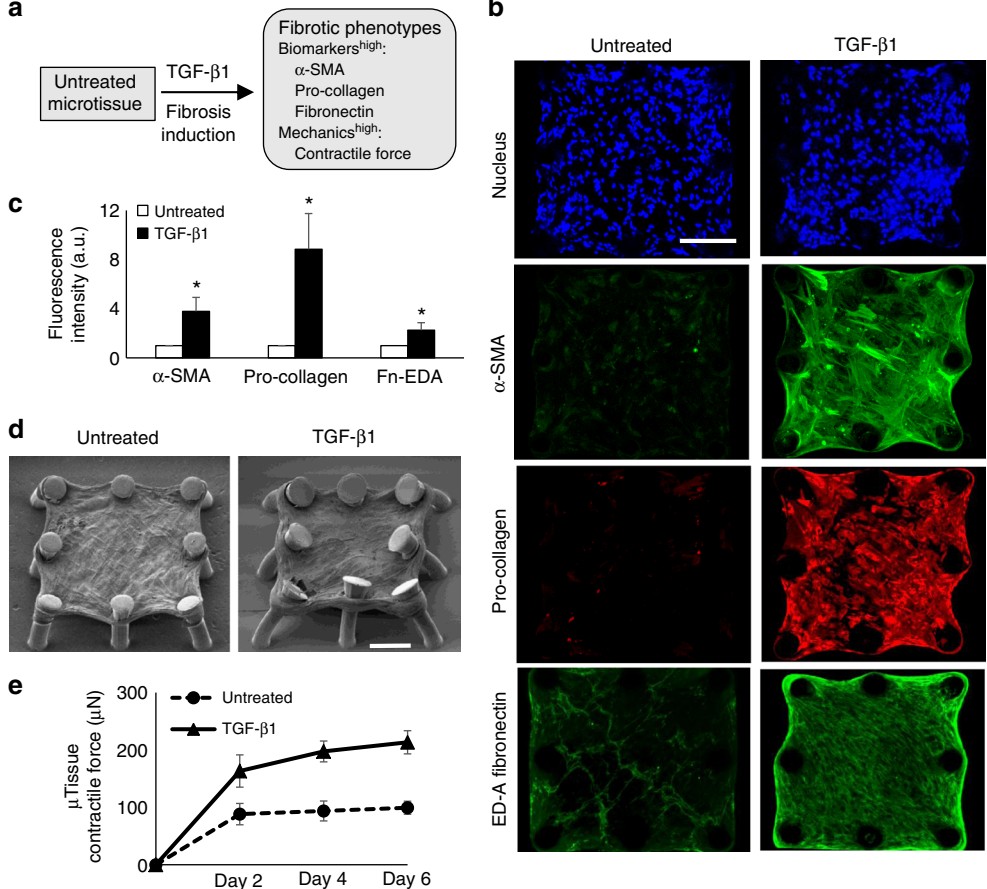

**Fig. 2** Recapitulation of tissue fibrogenesis in lung microtissues. **a** Overview of the strategy for fibrosis induction and evaluation based on the measurements of biomarker expression and tissue contractile force. **b** Continuous TGF-β1 treatment induced strong expressions of α-SMA stress fibers, cytosolic pro-collagen, and EDA-Fibronectin (Fn) in lung fibroblast-populated microtissue, as illustrated by representative fluorescent confocal images. **c** Fluorescence intensity levels of α-SMA, pro-collagen, and EDA-Fibronectin in TGF-β1-treated and untreated microtissues. **d** SEM images of an untreated and a TGF- β1-treated microtissue. Significant micropillar deflection caused by elevated tissue contraction can be seen in TGF- β1-treated microtissue. **e** Time-lapsed microtissue contractile force measurement. Contractile force of TGF-β1-treated samples nearly doubled that of untreated samples at every time point over a 6 day period. Data are reported as the mean ± SD. $n \geq 10$; *$P < 0.001$ when compared to untreated condition by one-way ANOVA with Tukey test. Scale bar is 200 μm

**Recapitulation of tissue fibrogenesis in lung microtissues.** Progression of fibrosis in lung interstitial tissue involves tissue stiffening and fibroblast activation into contractile myofibroblasts, characterized by expression of smooth muscle actin (α-SMA). Since TGF-β1 has been shown to play a key role in the development of fibrosis[5,27,28], we tested whether TGF- β1 induces fibrosis in lung fibroblast-populated and SAECs-populated microtissues (Fig. 2a). Continuous TGF-β1 treatment for 6 days induced strong expressions of α-SMA stress fibers and cytosolic pro-collagen and substantial increase in the deposition of highly aligned EDA fibronectin fibers in fibroblast-populated collagen microtissues (Fig. 2b). Measured fluorescence intensities of α-SMA, cytosolic pro-collagen, and EDA fibronectin in TGF-β1-treated condition were approximately two to eight times higher than those in untreated condition (Fig. 2c). These phenotypic changes were associated with significant increase in the contractile force generated by fibroblast-populated microtissues, as demonstrated by micropillar deflections (Fig. 2d; Supplementary Fig. 5). Time-lapsed force measurement showed that microtissue contractile force continuously increased under TGF-β1 treatment to around 210 μN over a 6 day culture period. In contrast, the force developed by untreated samples plateaued after 2 days at around 95 μN (Fig. 2e). Together, these measurements indicated the pathological transition of fibroblast-populated collagen

microtissue to a contractile state as a result of myofibroblast differentiation.

In SAECs-populated collagen microtissues, we did not observe significant increase in either the expression of biomarkers or the contractile force in TGF-β1-treated samples (Supplementary Figs. 6, 7), suggesting that airway epithelial cells are less responsive to TGF-β1 induction[29]. Next, we tested the effects of different ECM proteins on the fibrosis induction of the microtissues. Microtissues were fabricated using collagen, mixed ECM of collagen and fibrin and mixed ECM of collagen and matrigel, all populated with lung fibroblasts. There was no significant difference in either the expression of biomarkers or the contractile force between different ECM compositions with or without TGF-β1 treatment (Supplementary Figs. 8, 9). Together, these results showed that lung fibroblast-populated collagen microtissue is a compositionally simple yet effective model for lung interstitial fibrosis, and thus we chose this model for the study of the efficacy of the anti-fibrosis drugs.

Significantly increased lung tissue stiffness, often manifested by reduced lung interstitium compliance, is a major clinical indicator and outcome of pulmonary fibrosis[5]. Reducing or halting the decline in lung tissue compliance is used to evaluate efficacy of anti-fibrosis drugs[30,31]. To characterize these mechanical properties of our microtissues, we measured compliance and stiffness

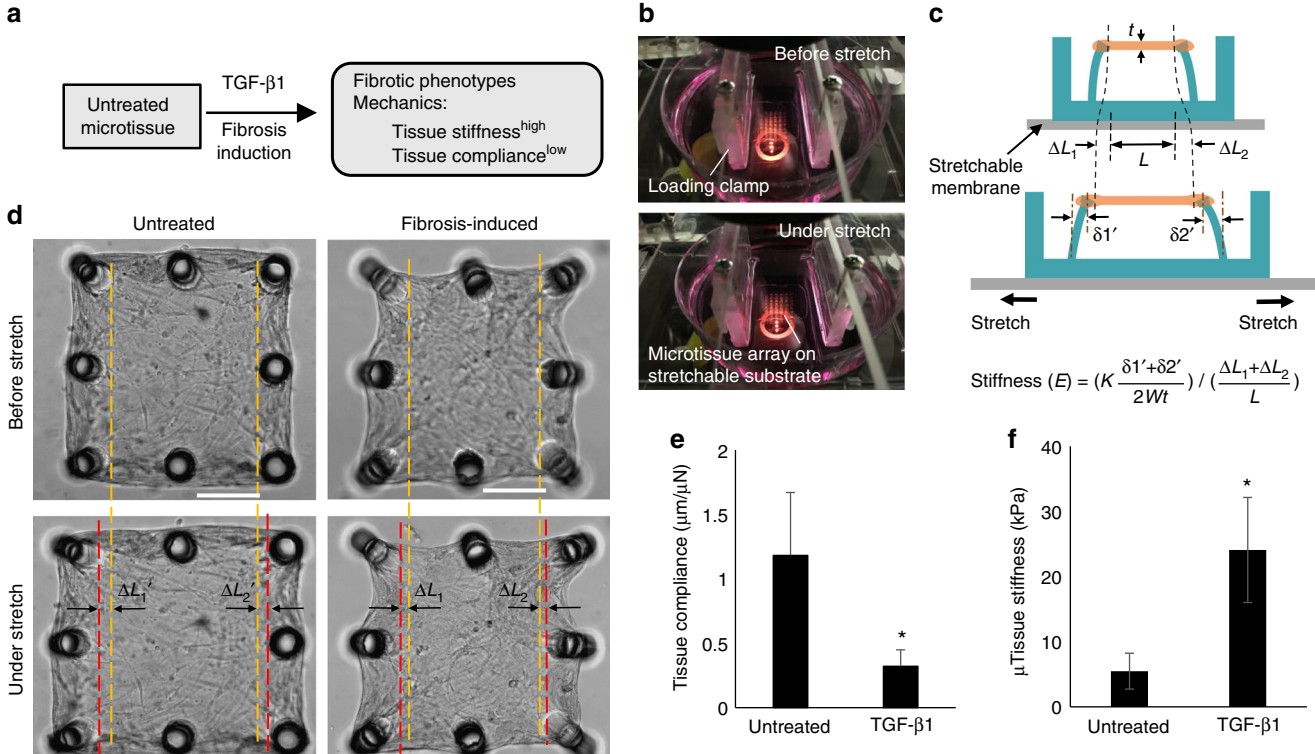

**Fig. 3** Modeling fibrotic tissue stiffening in lung microtissues. **a** Overview of the strategy for fibrosis induction and evaluation based on the measurement of tissue stiffness and compliance. **b** Mechanical stretching of the membranous microtissue mimics the respiratory distention of the alveolar sac walls. Top: unstretched microtissue array was bonded to a transparent stretchable substrate, which was mounted on a loading frame directly above microscope objective. Bottom: microtissue array under stretch. **c** Schematic shows the principle of tissue stiffness measurement. **d** Phase contrast images of microtissues before and under stretching. Stretch-induced extension of TGF-β1-treated microtissue ($\Delta L_1 + \Delta L_2$) was much less than that of untreated microtissue ($\Delta L_1' + \Delta L_2'$), indicating reduced compliance for TGF-β1-treated samples. **e** Plot of microtissue compliance shows a substantial decline in the compliance in TGF-β1-treated microtissues as compared to untreated microtissues. **f** Plot of microtissue stiffness shows much higher stiffness developed in TGF-β1-treated microtissues as compared to untreated microtissues. Data are reported as the mean ± SD. $n \geq 10$; *$P < 0.001$ when compared to untreated condition by one-way ANOVA with Tukey test. Scale bar is 200 μm

using mechanical stretching-enabled tensile tests (Fig. 3a). Stretching was performed in microtissue arrays casted on a thin, transparent silicone substrate to be stretched on a motor-driven loading frame (Fig. 3b; Supplementary Fig. 10). Owing to its membranous morphology, the microtissue deformed uniformly across its surface under stretching, mimicking the respiratory distention of lung interstitial tissues[32]. In tensile tests, microtissue strain was calculated as the tissue extension ($\Delta L_1 + \Delta L_2$) along the stretch direction divided by original tissue length ($L$), and tensile force was calculated based on the micropillar deflection caused by stretching ($F = k (\delta 1' + \delta 2')$). Elastic modulus ($E$, stiffness) of the microtissue was calculated as tensile stress divided by tensile strain (Fig. 3c). Microtissue compliance was calculated as stretching-induced tissue extension divided by tensile force (Supplementary Fig. 11). As shown in Fig. 3d, induced extension of TGF-β1-treated microtissues ($\Delta L_1 + \Delta L_2$) was less than that of untreated microtissues ($\Delta L_1' + \Delta L_2'$), indicating reduced compliance for TGF-β1-treated samples (Fig. 3e). While the stiffness of untreated microtissues was found to be 5.5 ± 2.8 kPa (mean ± SD), matching that of healthy lung tissue[33,34], the stiffness of TGF-β1-treated tissues (24.0 ± 8.0 kPa) was substantially higher and matched with those of human and mouse fibrotic lung tissues (Fig. 3f)[33,34]. Hence, these alterations in the cellular phenotype, ECM protein expression, and tissue mechanics reproduced features of lung fibrosis in the human lung microtissues.

**Modeling the biomechanics of traction bronchiectasis.** In fibrotic lung, traction bronchiectasis occurs when small bronchial

openings dilate due to traction force exerted by the surrounding fibrotic tissue (Fig. 4a)[23,35]. To model this unique fibrosis feature of the lung, we patterned stress concentrations to induce the dilation of tissue openings in our engineered microtissues (Fig. 4b). We first explored microtissue designs that will create regions with high level of stress concentration. Using FE models, we compared two square microtissue designs, supported on either flexible (Fig. 4c) or rigid micropillars (Fig. 4d). Increased micropillar rigidity restricted microtissue contraction on the edges and thereby significantly increased the level of stress concentration around micropillars (Fig. 4c, d; Supplementary Fig. 12). To further increase local stress levels, we designed a long microtissue supported on rigid micropillars. The unique geometry of the long microtissue with a high $x/y$ aspect ratio of 8 resulted in the formation of a high stress band in the belly of the microtissue (Fig. 4e; Supplementary Fig. 12). The uniaxial stress filed in this band induced the dilation of openings in the microtissue belly. In contrast, artificially created wound has been found to heal in square microtissues where biaxial stress field dominates[20]. To experimentally model the mechanical process of traction bronchiectasis, lung fibroblast-populated long microtissue was fabricated and then induced to fibrosis using TGF-β1 treatment (Fig. 4f). Continuous TGF-β1 treatment for 6 days induced expression of α-SMA stress fibers across the entire microtissue in addition to dilated tissue openings around the micropillars and in the tissue center. However, these effects were absent in untreated conditions (Fig. 4g; Supplementary Fig. 13). Such differences observed in TGF-β1-treated and untreated

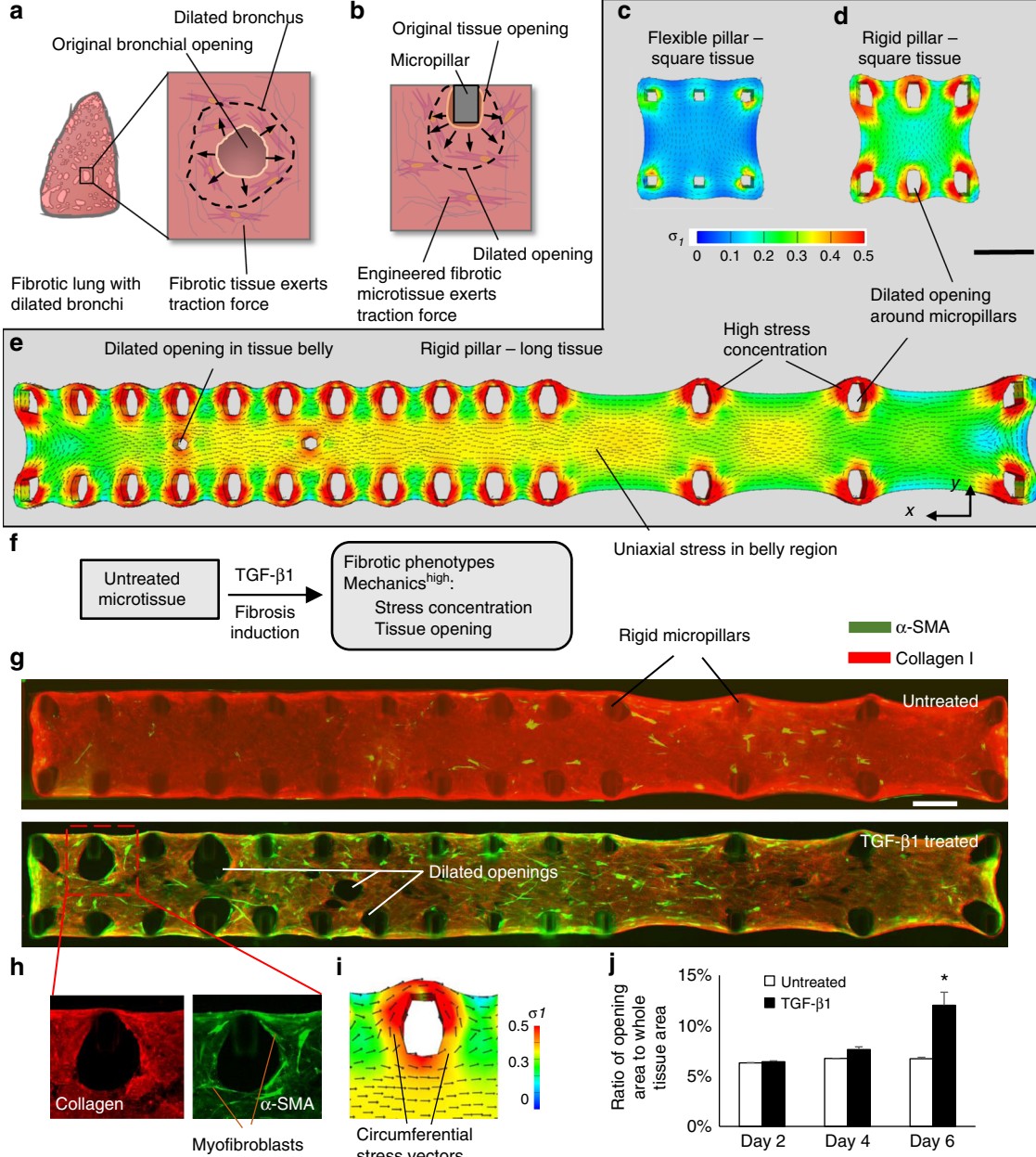

**Fig. 4** Modeling the biomechanics of traction bronchiectasis. **a** Schematic shows the formation of numerous cystic airspaces in the fibrotic lung interstitium due to traction force-induced bronchial dilation. **b** The bronchial dilation was modeled through inducing the dilation of tissue openings in engineered fibrotic microtissues. FE simulated first principal stress distribution of a square microtissue supported by flexible micropillars (**c**), a square microtissue supported by rigid micropillars (**d**) and a long microtissue supported by rigid micropillars (**e**). Note high stress concentration around the micropillars in **d** and **e** induced dilation of the tissue opening. **f** Overview of the strategy for fibrosis induction in long microtissue and fibrosis evaluation based on the measurement of stress concentration and tissue opening size. **g** Merged immunofluorescence images of α-SMA and collagen type-I of untreated and TGF-β1-treated long microtissues. Apparent dilation of openings around micropillars and in the belly region can be observed in TGF-β1-treated condition. Scale bar is 500 μm. **h** Enlarged views of collagen type-I and α-SMA of highlighted region in **g**. α-SMA positive myofibroblasts aligned circumferentially around the dilated openings, matched well with the direction of simulated principal stress vectors (**i**). **j** Plot of the percentage of microtissue area occupied by the openings. The opening area of TGF-β1-treated sample is significantly larger than that of untreated sample at day 6. Data are reported as the mean ± SD. $n \geq 5$; *$P < 0.001$ when compared to untreated condition by one-way ANOVA with Tukey test

conditions was predicted by FE models of identical geometry but assigned with different levels of tissue contractility (Supplementary Fig. 13), suggesting that fibrosis-associated high tissue contractility played a critical role in the formation of tissue openings. This point was further validated by the matching distribution of contractile myofibroblasts (Fig. 4h) and simulated principal tensile stresses (Fig. 4i) around the tissue openings. Quantitative measurement of microtissue area showed that tissue opening area

in TGF-β1-treated sample is significantly larger than that in untreated sample at day 6 (Fig. 4j). Therefore, we propose to use this measurement as a phenotypic parameter for the efficacy analysis of anti-fibrosis drugs.

**Predicting anti-fibrosis drug efficacy using microtissue model.** We next tested the suitability of our novel human lung

microtissue array as fibrosis disease models for predicting the efficacy of anti-fibrosis drugs in humans. Pirfenidone and Nintedanib were selected for proof-of-concept as they are the only anti-fibrosis drugs approved by FDA to treat IPF; both drugs were shown to reduce the decline in lung compliance in IPF patients[30,31]. We performed both preventative and therapeutic treatments on diseased microtissue model using these drugs at commonly used in vitro concentrations (Supplementary Fig. 14). In preventative treatment, anti-fibrosis drugs were co-administered with TGF-β1 at the beginning of microtissue culture and remained throughout the entire culture period (Fig. 5a). In therapeutic treatment regimens, TGF-β1-induced fibrosis was allowed to progress for the first three days and anti-fibrosis drugs were then administered and remained to the end of experiment.

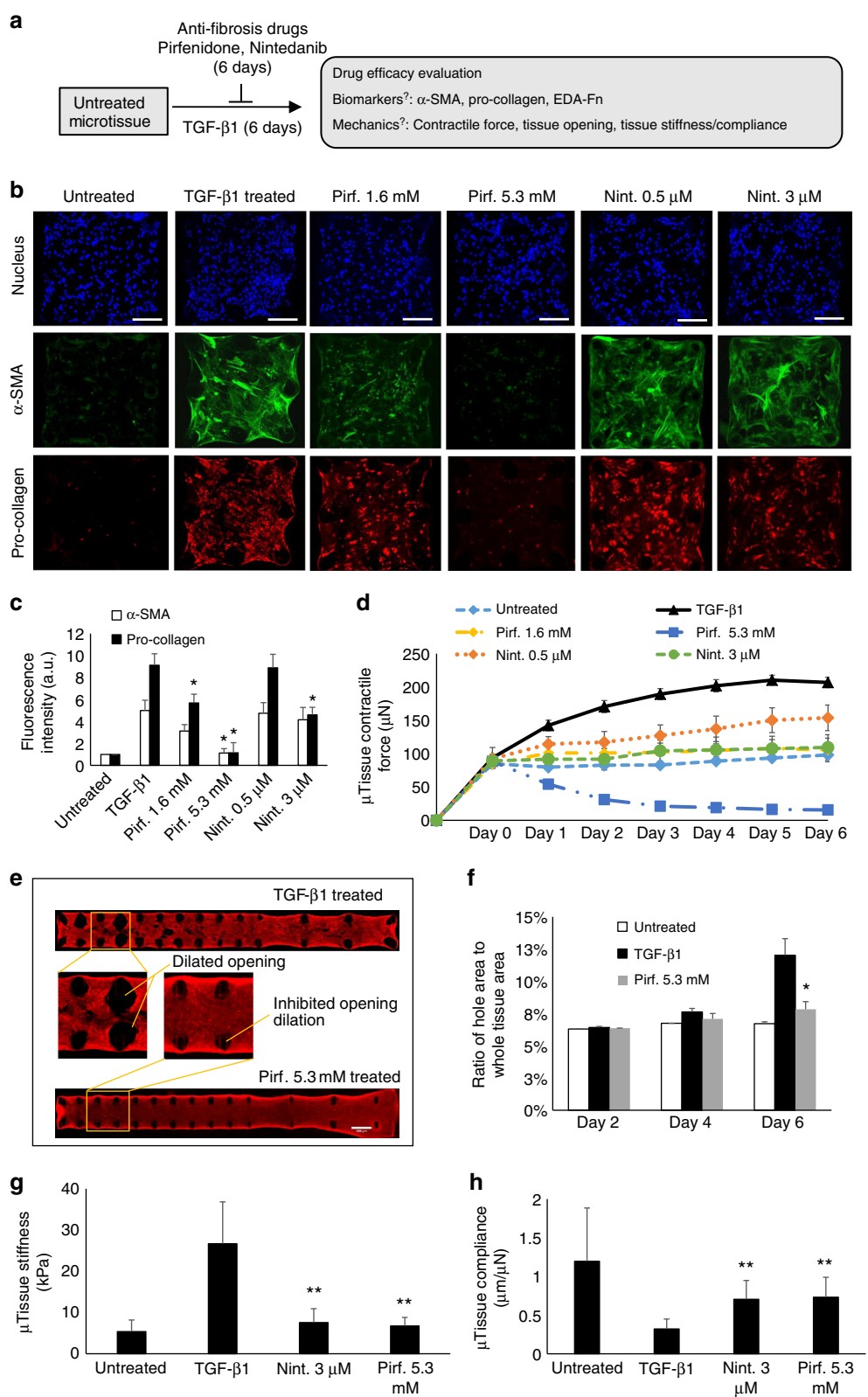

Pirfenidone and Nintedanib inhibited myofibroblast activation in a preventative anti-fibrosis treatment regimen to different degrees. Expressions of α-SMA, pro-collagen, and EDA Fn were increasingly inhibited by increasing pirfenidone concentrations with maximum effect at 5.3 mM (Fig. 5b, c; Supplementary Fig. 15). In contrast, Nintedanib inhibited the stress fiber assembly but not fluorescence intensity of α-SMA at all concentrations, and only high Nintedanib concentration (3 μM) inhibited pro-collagen and EDA Fn expressions (Fig. 5b, c; Supplementary Fig. 15). Only pirfenidone at 5.3 mM was found to modestly affect cell proliferation (Supplementary Fig. 16).

Pirfenidone in high dosage (5.3 mM) almost entirely abolished tissue contractility after 3 days culture in time-lapsed tissue contractile force measurements (Fig. 5d). Loss of tissue contractility was associated with severely reduced collagen ECM compaction, as demonstrated by the microtissue area that is larger than that of untreated samples (Supplementary Fig. 17a). Low dosage pirfenidone (1.6 mM) was overall less effective but was able to inhibit TGF-β1-induced force increase and maintain the tissue force at a level similar to untreated sample (Fig. 5d). Compared to pirfenidone, the inhibitory effect of Nintedanib on contractile force was moderate (Fig. 5d), consistent with its limited inhibition of TGF-β1-induced α-SMA expression in myofibroblasts. Despite the moderate inhibition on tissue contractile force, Nintedanib at 3 μM was able to substantially inhibit the compaction of collagen ECM, as demonstrated by the microtissue area that is larger than that of TGF-β1-treated samples (Supplementary Fig. 17a). Since Nintedanib has been shown to also act on the platelet-derived growth factor (PDGF) pathway[36], we induced microtissue contraction using PDGF with simultaneous addition of Nintedanib. Nintedanib strongly inhibited PDGF induced tissue contractility (Supplementary Fig. 18), demonstrating the power of the assay to discriminate between drug actions on different fibrosis pathways.

Next, we performed preventive anti-fibrosis treatments in long microtissues to study drug effects on traction bronchiectasis-like tissue opening formation. Pirfenidone at 5.3 mM almost completely inhibited TGF-β1-induced opening dilation (Fig. 5e, f; Supplementary Fig. 19). This pirfenidone dosage also strongly inhibited TGF-β1-induced tissue stiffness increase and compliance decrease, and maintained tissue stiffness and compliance comparable to untreated samples. Similar inhibitory effects on tissue stiffness/compliance was also observed on Nintedanib at 3 μM (Fig. 5g, h; Supplementary Fig. 20).

For therapeutic drug treatments, i.e., treatments after onset of fibrosis, we treated microtissues with TGF-β1 for 3 days, followed by drug treatment for another 3 days in the absence of TGF-β1 (Fig. 6a). Removing TGF-β1 in the later 3 days alone (TGF-β1 +3/−3) had no effect on already-established α-SMA expression, pro-collagen expression (Fig. 6b, c), EDA Fn expression (Supplementary Fig. 21), tissue contractility (Fig. 6d), or tissue stiffness (Fig. 6e) as compared to continuous TGF-β1 treatment for 6 days. Pirfenidone at 5.3 mM but not at 1.6 mM significantly

reduced the expression of α-SMA, pro-collagen and EDA Fn (Fig. 6b, c; Supplementary Fig. 20), tissue contractility (Fig. 6d), and collagen ECM compaction (Supplementary Fig. 17b). Nintedanib at 3 μM moderately reduced pro-collagen expression but did not reduce the expression of α-SMA and EDA Fn (Fig. 6b, c; Supplementary Fig. 21), tissue contractility (Fig. 6d) or collagen ECM compaction (Supplementary Fig. 17b). Pirfenidone (5.3 mM) treatment also strongly reduced tissue stiffness by ~50% after 3 days of therapeutic treatment (Fig. 6e). As a result, fibrosis-induced decline in tissue compliance was rescued, and compliance increased from 0.35 μm/μN at day 3 to 0.66 μm/μN at day 6 (Fig. 6f). Together, these data showed the reversal of fibrosis by pirfenidone treatment. We also observed 17% reduction in tissue stiffness from day 3 to 6 by Nintedanib at 3 μM. Correspondingly, the compliance increased from 0.35 μm/μN at day 3 to 0.41 μm/μN at day 6 (Fig. 6e, f).

## Discussion

Fibrosis is one of the leading causes of death in developed countries, but effective medical therapies remain elusive[30,31]. Even though several anti-fibrotic drug candidates, such as TGF-β inhibitors, αvβ6 integrin blockers, IL-4 and IL-13 antibodies, LoxL2 inhibitors and microRNAs, have been identified, the clinical translation is slow due to the lengthy and costly nature of the clinical trials[1,2]. The approval of Pirfenidone and Nintedanib is a major progress in anti-fibrotic treatment, but these drugs are only approved to treat IPF and the cost is high. Furthermore, the mode of action of these drugs is not fully understood[3,37,38]. Therefore, continued research and development of anti-fibrotic treatments, including the screening and evaluation of new drugs and repurposing of the existing drugs, is urgently needed. Here we provide a human lung fibrotic microtissue array that allows multi-parameter, phenotypic screening of the anti-fibrosis drug candidates in a relatively high-throughput manner. The adoption of the microtissue array format offers orders of magnitude scale-up advantages over tissue biopsy samples and conventional engineered tissues[10,11], leading to improved experimental throughput. Further integration of the microtissue array with multi-well plate enabled parallel analysis of multiple drug conditions. In this system, all the measurements including those performed on fluorescence intensity, tissue mechanical properties and tissue opening formation were image-based and were performed using a conventional epi-fluorescence microscope. The easy readout of multiple physiologically relevant parameters from a large sample population under a variety of biochemical treatments makes this system well suited for the application of high-content drug screening[9].

Tissue fibrosis progresses along with substantial changes in the tissue mechanics. While broadly used planar 2D cell assays can be configured into high-throughput, high-content format and are useful in the study of the molecular mechanisms of the anti-fibrosis drugs[7,36], they do not recapitulate the biomechanical

---

**Fig. 5** Anti-fibrosis drug efficacy under preventative treatment. **a** Overview of the strategy for preventative anti-fibrosis treatment and evaluation of the anti-fibrosis efficacy based on the measurement of biomarker expression and tissue mechanical properties. Pirfenidone (Pirf.) and Nintedanib (Nint.) were co-administered with TGF-β1 at the beginning of experiments and remained throughout the 6 day treatment period. **b** Representative immunofluorescence images of nuclei, α-SMA and pro-collagen of microtissues at day 6, with or without preventative anti-fibrosis treatments. Scale bar is 200 μm. **c** Plot of tissue-level fluorescence intensity of α-SMA and pro-collagen at day 6. **d** Time-lapsed measurement of microtissue contractile force. **e** Representative fluorescent images of collagen type-I of TGF-β1-treated and Pirf.-treated long microtissues. Zoom-in views showed that dilation of opening was inhibited by Pirf. treatment. Scale bar is 500 μm. **f** Time-lapsed plot of the percentage microtissue area occupied by the tissue openings. Pirfenidone treatment almost completely inhibited opening dilation at day 6. **g** Plot of the microtissue stiffness measured by tensile test at day 6. **h** Plot of the microtissue compliance measured at day 6. Data are reported as the mean ± SD. $n \geq 10$; *$P < 0.05$ when compared to TGF-β1-treated condition; **$P < 0.001$ when compared to TGF-β1-treated condition. Statistical significance was determined by one-way ANOVA with Tukey test

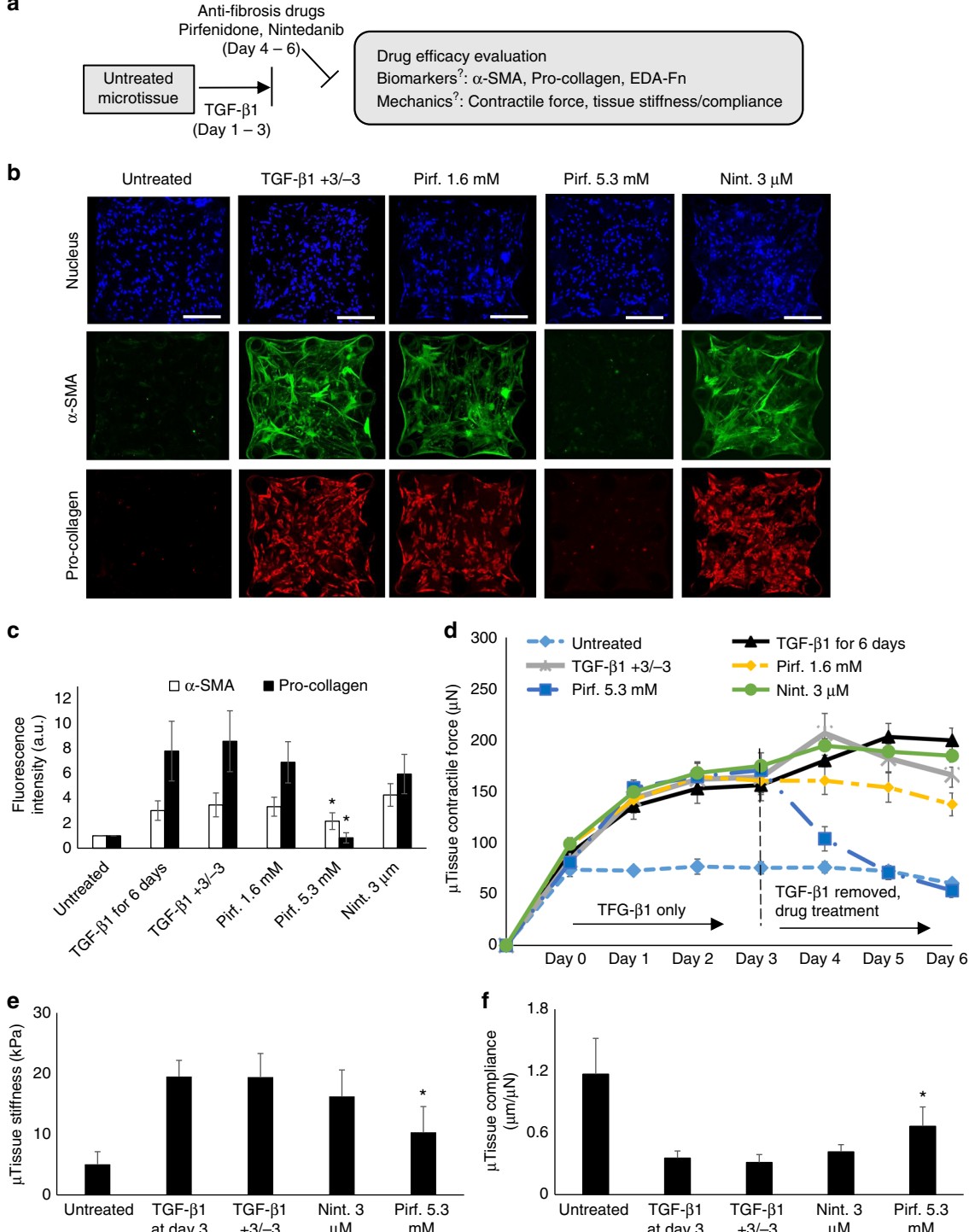

**Fig. 6** Anti-fibrosis drug efficacy under therapeutic treatment. **a** Overview of the strategy for therapeutic anti-fibrosis treatment and evaluation of the anti-fibrosis efficacy based on the measurement of biomarker expression and tissue mechanical properties. Initial fibrosis progression was induced using TGF-β1 in the first three days and anti-fibrosis treatments were applied from day 3 to 6 without the presence of TGF-β1. **b** Representative immunofluorescence images of nuclei, α-SMA and pro-collagen of microtissues at day 6, with or without therapeutic anti-fibrosis treatments. TGF-β1 +3/−3 represents 3 days of TGF-β1 treatment followed by 3 days of normal culture condition without TGF-β1. Scale bar is 200 μm. **c** Plot of tissue-level fluorescence intensity of α-SMA and pro-collagen at day 6. **d** Time-lapsed measurement of microtissue contractile force. **e** Plot of microtissue stiffness measured by tensile test at day 6. **f** Plot of microtissue compliance measured at day 6. Data are reported as the mean ± SD. $n \geq 10$; *$P < 0.05$ when compared to TGF-β1-treated condition. Statistical significance was determined by one-way ANOVA with Tukey test

changes occurred in the fibrotic tissue and organ. The contraction of fibroblast/myofibroblast spheroids has been used as a phenotypic parameter in anti-fibrosis screening[8,39]; however, the lack of morphological control and stiffness measurement of this model resulted in limited prediction power and accuracy for drug efficacy. Tissue slices prepared from different organs allowed the study of anti-fibrosis drug efficacy on multiple cell types in well-retained anatomical structures, but the low-throughput of this model limited its application in drug screening[40,41]. The fibrotic lung microtissue array system presented in the current study recapitulated key biomechanical properties of both healthy and fibrotic lung alveolar tissues, allowing for simultaneous biochemical and biomechanical analysis of the efficacy of the anti-fibrosis drugs; therefore, it represents a novel approach for drug screening with improved physiological relevance, accuracy and throughput. Due to the similar pathology between different fibrotic diseases, the current fibrotic microtissue array system can potentially be used to model the fibrotic disease in other organ systems, such as liver cirrhosis, renal fibrosis and cardiac fibrosis, and help to develop the anti-fibrosis treatment for these diseases.

We tested the utility of this system for the prediction of anti-fibrosis drug efficacy and showed results of Pirfenidone and Nintedanib as a proof of principle. We showed that while these two drugs displayed different degrees of inhibition of myofibroblast activation and tissue contractility, they both can effectively inhibit tissue stiffening and the decline in tissue compliance through potentially preventing cell-mediated ECM strain stiffening. Previous studies on reconstituted cell-laden collagen/fibrin gels have shown that cell contraction pulls out thermal bending undulations of the ECM fibril meshwork to cause meshwork compaction and gel stiffening[42,43], a phenomenon broadly known as strain stiffening[18,44,45]. In this study, using microtissue area as a measurement of ECM compaction, we showed strong and moderate inhibition of TGF-β1-induced microtissue contractility and ECM compaction by preventative treatment with pirfenidone at 5.3 mM and Nintedanib at 3 μM, respectively. In both cases, tissue stiffening was inhibited. These data suggest that a threshold ECM compaction level, which is similar to that of untreated samples, is critical to the inhibition of microtissue stiffening, and tissue stiffening can be prevented if the microtissue area is maintained equal to or larger than that of the untreated sample (Supplementary Fig. 17). This finding started to unveil the potential biomechanical mechanism of action of Pirfenidone and Nintedanib on fibrosis treatment. Future studies on tissue mechanics such as tissue microstructure under the treatment of these two drugs and other anti-fibrosis drugs will help to deepen the understanding of the mode of action of anti-fibrosis treatments.

The high span length to thickness ratio ($S/t$ ratio) of the presented microtissues is critical to the modeling of the biomechanical properties of both healthy and fibrotic lung interstitial tissues. Like other cell-laden ECM hydrogels[42,43], a microtissue gains its stiffness through cell-contraction-mediated ECM compaction. During microtissue formation, the membraneous tissue morphology formed because tissue compacted freely through its thickness, but its compaction in the $x$–$y$ plane was restricted by the micropillars. In untreated condition, this resulted in a relatively loosely compacted microtissue whose low tissue stiffness (~5 kPa) matched to that of the in vivo lung tissue[33,34]. But under TGF-β1 treatment, high contraction induced further tissue compaction, as demonstrated by the reduced microtissue area (Supplementary Fig. 17), resulted in the stiffness increase to around 20 kPa, matching well with that of the fibrotic lung tissue[33,34]. In the long microtissue that models the biomechanics of traction bronchiectasis, the thin, membraneous morphology of the

microtissue allowed easy dilation of tissue openings under myofibroblasts-generated traction forces. The stretching of the membraneous tissue, which mimics forced alveolar distention[32], enabled mechanical tensile tests on the microtissue. These tests allowed for the measurement of the stiffness and compliance of the microtissues, which are major biomechanical parameters used to assess the therapeutic efficacy of the anti-fibrosis drugs. Owing to motorized control of the stretching system, it can be easily adopted to apply long-term, cyclic stretching to the microtissue array. Since cyclic loading has been shown to increase the biomechanical activity of lung cells in 2D culture[7,8], its application in microtissue may shed new light on the study of anti-fibrosis therapies. Furthermore, the current microtissue system can be used to model the physiology and pathology of other membrane-like tissue types, such as the fibrosis in the retina and intestinal fibrosis associated with Crohns disease. Owing to the high level of stress concentration generated around micropillars, currently presented microtissue model for drug screening is limited to medium-size and simple geometry. Future design improvements that reduce stress concentration, potentially through increasing the contact area between tissue and PDMS supports, could allow the design and construction of microtissue models with more complex geometry.

In conclusion, we have leveraged the unique remodeling capability of suspended microtissues to develop a fibrotic microtissue array system that recapitulates the key aspects of fibrogenesis in lung interstitial tissues. We utilized this system for predicting the efficacy of anti-fibrosis drugs and showed the results of Pirfenidone and Nintedanib as a proof of principle. We showed that while these two drugs acted differently on the fibrotic cells, they are both effective in restoring the fibrotic tissue compliance. Such findings across different length scales from cells to organ started to unveil the potential mode of action of these anti-fibrosis drugs. It is expected that the multiscale and pathophysiologically relevant modeling capability of the fibrotic microtissue system will expedite the translation of anti-fibrotic therapies from the laboratories to the clinics. Future work that includes diseased lung cells such as those isolated from IPF patients, vascular cells such as pericytes, immune cells, and other ECM components such as elastin may shed new light on the cell–ECM interaction involved in fibrosis and further expand the physiological relevancy of the current system.

## Methods

**Cell culture**. Primary normal human lung fibroblasts (NHLFs) were purchased from Lonza and were maintained in manufacturer supplied growth medium (FGM-2 BulletKit, CC3132, Lonza). NHLFs were cultured up to six passages with media change every 3 days. Human primary SAEC (PCS-301-010, ATCC) were maintained in manufacturer supplied growth medium (PCS-300-030 and PCS-300-040, ATCC) supplemented with 100 U/ml penicillin, and 100 μg/ml streptomycin.

**FE modeling of microtissue formation**. FE model of the microtissues with various geometries was constructed in FEBio[46]. Microtissue geometry was discretized by 3D quadratic tetrahedral elements capable of large deformation. The active contraction of the cell population was represented by a solid mixture constitutive material consisting of a neo-Hookean solid component and an isotropic contractile stress component[26]. The compressive neo-Hookean solid allows compaction of the microtissue model under active contraction. In the comparison between four-leaflet microtissue and single-leaflet microtissues, individual micropillars were considered to be flexible (Fig. 1); but in the comparison between long microtissue and square microtissues, micropillars in the long microtissue and rigidly-supported square microtissue were considered to be rigid (Fig. 4). Simulated first principal stress contour was plotted over deformed model geometry in the results, with overlaid plot of stress vectors, where applicable. To compare the stress level in different microtissue designs, the histograms of the first principal stress were plotted and compared.

**Micropillar device fabrication and microtissue seeding**. The layout of the micropillar array was designed based on FE models. Micropillar arrays were fabricated using a multi-layer microlithography technique similar to that described

previously[18,19]. Briefly, SU-8 masters were created through spin coating, alignment, exposure and baking of multiple layers of SU-8 photoresists. The cross-sectional difference between the micropillar head and leg sections was achieved through depositing a thin layer of SU-8 doped with S1813 that served to prevent UV light penetration to the leg section. UV exposure was performed on an OAI maskaligner with a U-360 band pass filter. PDMS (Sylgard 184, DowCorning) stamps were casted over the SU-8 master at 10:1 mixing ratio. Micropillar devices were then produced through replica molding from stamps in P35 petridishes or 12 well plate. Microtissue seeding was performed following a protocol previously established[17,18]. Briefly, micropillar device was sterilized and treated with Pluronic F-127 (P2443, Sigma) to prevent non-specific cell adhesion to the PDMS surface. NHLFs or SAECs cells were mixed with collagen type-I (rat tail, Corning) at a final concentration of 3 mg/ml, and the mixture was introduced into the microwells through centrifugation. Collagen solution was crosslinked and then maintained in appropriate growth media in a $CO_2$ incubator. For the purpose of testing the combination of ECMs, NHLFs were mixed with either a ECM mixture containing neutralized collagen (3 mg/ml) and 20% v/v Matrigel (356231, CORNING) or a ECM mixture containing neutralized collagen (3 mg/ml) and 2.5 mg/ml human Fibrinogen (FIB1, Enzyme Research Laboratories). The cell-laden ECM mixture was then seeded and polymerized in a manner similar to cell-laden collagen.

**Pharmacological treatment.** Anti-fibrotic drugs Pirfenidone (P1871, TCI America) and Nintedanib (S1010, Selleckchem) were purchased from commercial suppliers. In pharmacological tests, F-12k media (Thermo Fisher) supplemented with 2% FBS, 100 U/ml penicillin and 100 μg/ml streptomycin was used as a base medium and negative control for NHLF-populated microtissues. To induce myofibroblast differentiation of the NHLFs, 5 ng/ml of TGF-β1 (T5050, Sigma) was added to the base media and maintained for different durations. Pirfenidone at 1.6 and 5.3 mM and Nintedanib at 0.5 and 3 μM were used in two treatment approaches. For direct comparison, the high concentrations we selected for both drugs are approximately 50 times of their concentrations in human plasma[47]. These concentrations were broadly used in previous in vitro studies and are not toxic to cultured cells[36,38,47]. The absorption of drugs by PDMS is negligible (Supplementary Note 1). In preventative treatment, anti-fibrosis drugs were co-administered with TGF-β1 at the beginning of microtissue formation and maintained for 6 days, and in therapeutic treatment TGF-β1-induced fibrosis was allowed to progress for the first 3 days and TGF-β1 was removed, and anti-fibrosis drugs were then administered and remained for another 3 days. In SAEC-populated microtissues, manufacturer supplied basal medium (PCS-300-030, ATCC) supplemented with 1% FBS, 100 U/ml penicillin, and 100 μg/ml streptomycin was used as a base medium and negative control. 5 ng/ml of TGF-β1 was added to the base media for the fibrosis induction of SAEC-populated microtissues. Live/dead assay was performed to examine the effects of anti-fibrosis drugs on lung fibroblast viability.

**Microtissue contraction force measurement.** The spontaneous contraction force generated by individual microtissues was determined from the deflection of micropillars based on cantilever bending theory (Supplementary Fig. 5). Micropillar deflection was measured using phase contrast microscopy, by comparing the deflected position of the centroid of each pillar top to the centroid of its base. Microtissue contraction force was measured for the same batch of samples on a daily basis for each pharmacological condition. Image analysis was performed in ImageJ and the absolute values of calculated force from each micropillar were added up and divided by 2 to reflect the collective force generated by one microtissue. Micropillar's spring constant, which is 0.66 μN/μm for flexible micropillars in four-leaflet microtissue and single-leaflet square microtissues and 1.99 μN/μm about $x$ axis and 7.94 μN/μm about $y$ axis for rigid micropillars in long microtissues, was determined based on the actual geometry of the micropillar and the elastic modulus of the PDMS.

**Stretching of microtissue array.** A mechanical stretching system has been developed to stretch membranous lung microtissues, which mimics the respiratory motion of lung interstitial tissues. A transparent stretchable silicone substrate (SMI silicone sheeting, 0.01-inch NRV G/G 40D, Saginaw, MI) was mounted on a custom-made loading frame driven by a DC motor controlled by an Arduino microcontroller. The stretchable substrate was fixed at one end and was allowed to move at the other end. By controlling the displacement of the moving end, we were able to achieve single step stretching or cyclic stretching of the microtissue device. The mounting fixture allows the silicone substrate to be lowered to close to the bottom of a P100 petri-dish, enabling real-time measurement of the microtissue deformation and micropillar deflection of multiple microtissues through optical imaging. Before stretching, empty microtissue array device was casted on the silicone substrate, and microtissue seeding and culture were performed following the same procedure as described in earlier sections. All microtissues in each $8 \times 8$ array were stretched simultaneously, which represents a significantly higher throughput than conventional tissue stretching experiments. The tensile strain transfer from the silicone substrate to microtissue and the strain variation across different microtissue samples were characterized, and consistent tensile strain has been found in individual microtissues across different regions of the device.

**Microtissue stiffness and compliance measurement.** Tensile tests were performed using the stretching system on untreated, TGF-β1-treated and anti-fibrosis drug treated microtissues to measure their stiffness and compliance. After 6 days of culture, microtissue array-mounted silicone substrate was mounted on the stretching machine and was stretched to 120% of initial length. Phase images of the microtissue and the micropillars before and after tensile test were taken. Microtissue strain was calculated as the tissue extension ($\Delta L_1 + \Delta L_2$) along the tensile direction divided by original tissue length ($L$) (Fig. 3c). Stretching-induced tensile force was calculated based on the micropillar deflection caused by stretching ($F = k$ ($\delta 1' + \delta 2'$)) (Fig. 3c). Summation of the tensile forces from all micropillars was divided by 2 and then divided by the cross-section area of the microtissue to obtain the tensile stress. Elastic modulus ($E$, stiffness) was calculated as tensile stress divided by tensile strain. The thickness of microtissues was obtained using confocal microscopy. Microtissue compliance was calculated as stretching-induced tissue extension ($\Delta L_1 + \Delta L_2$) divided by tensile force ($F$).

**Immunofluorescence and image analysis.** Human lung cell-populated microtissues were fixed with 4% formaldehyde, permeabilized by cold methanol at −20 °C, blocked by 3% BSA, incubated with primary antibodies against α-smooth muscle actin (Abcam, ab7817, 1:300), pro-collagen type-I (Millipore, MAB1913-C, 1:400), EDA-Fibronectin (Abcam, ab6328, 1:400) and rat collagen type-I (Millipore, AB755P, 1:300), and labeled with fluorophore-conjugated, anti-IgG antibodies (Thermo Fisher, AlexaFluor, 1:400). The nuclei were counterstained with Hoechst 33342 (Thermo Fisher, 1:1000). To measure the fluorescence intensity of α-SMA and pro-collagen type-I in different microtissues under varied pharmacological conditions, images were taken on a Nikon Ti-U microscope equipped with a Hamamatsu ORCA-ER CCD camera using 10× Plan Fluor objective under identical imaging condition. Fluorescence intensity was measured in ImageJ by taking the integrated intensity of a region of interest bounded by the inner edge of micropillars and then subtracting the background intensity. Fluorescence intensity was normalized by the cell number in each microtissue. Confocal images of the microtissue were taken on an Andor Technology DSD2 confocal unit coupled to an Olympus IX-81 motorized inverted microscope equipped with Plan-Apochromat ×10 air objective. Optical slice of 1 μm was used for all channels. The stack of images was processed using the $Z$ stack tool in ImageJ to obtain the projected 2D views.

**Scanning electron microscopy.** The microdevices seeded with normal HLFs were fixed using 2% glutaraldehyde (ACROS organics) for 1.5 h. Next they were dehydrated through series of ethanol treatment at 15, 30, 50, 70, 90, and 100%. Finally, Hexamethyldisilazane (HMDS, Alfa Aesar) was added to the samples to prepare them for SEM. The SEM was done with the Hitachi SU70 Field Emission Scanning Electron Microscope (FESEM).

**Statistics.** Data are presented as the mean with error bars showing the standard deviation (SD). Data were analyzed by one-way ANOVA (Minitab 17 software). Significance difference was verified by the post-hoc Tukey method with certainty of $P < 0.05$ unless otherwise mentioned.

**Data availability.** All data supporting the findings of this study are available within the article and its supplementary information files or from the corresponding author upon reasonable request.

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

### Acknowledgements

Research reported in this study was supported by the National Institute of Biomedical Imaging and Bioengineering of the National Institutes of Health under award number R01EB019411 (R.Z.). The content is solely the responsibility of the authors and does not necessarily represent the official views of the National Institutes of Health. We would also like to acknowledge the funding support from the School of Engineering and Jacobs School of Medicine and Biomedical Sciences at the University at Buffalo and Buffalo Clinical and Translational Science Institute. Authors thank Samantha Bundy for assistance with scientific illustration.

### Author contributions

R.Z. conceived the idea; M.A., B.H., and R.Z. designed the experiments; M.A., Y.L., S.V., N.W., I.H., Z.C., and R.Z. performed experiments and analyzed data; M.A., B.H., and R. Z. wrote the manuscript; and R.Z. provided financial and administrative support.

### Additional information

**Competing interests:** The authors declare no competing interests.

