## [Peer Review File · Nature Communications]

Reviewers' comments:

Reviewer #1 (Remarks to the Author):

The manuscript "Fibrotic microtissue array to predict anti-fibrosis drug efficacy" by Asmani et al. aims to establish an in vitro model of lung fibrosis capable of supporting the evaluation of candidate therapeutic approaches for treatment of idiopathic pulmonary fibrosis (iPF) and other diseases. This avenue for investigation is of great interest given the intractable nature of fibrotic diseases and the absence of viable approaches that extend survival for iPF and radiation-induced PF patients. The manuscript describes a very interesting approach in which suspended cultured microtissues are capable of recapitulating key aspects of tissue stiffening and contraction, with approaches that attempt to model total lung capacity. Focus is given to two compounds that show promise in mitigating fibrotic tissue stiffening, Pirfenidone and Nintedanib, although neither has demonstrated a clear survival benefit clinically. Suspended constructs are evaluated for their mechanical properties in response to the fibrosis triggering agent TGF- β 1, and to subsequent treatment by the two abovementioned compounds. A mix of micropillar arrays and stretchable silicone elastomer films are used to assay biomechanical response to the various conditions studied. Overall, this appears to be an interesting and useful study that significantly expands upon the value of current alveolar disease models for fibrosis, such as the Huh / Ingber lung-on-a-chip, which has many drawbacks in this application space as the authors of the current manuscript rightly point out.

In spite of the interesting and useful results obtained and reported, the manuscript suffers from many significant flaws that would limit its utility and value to the literature unless they are rectified. These are noted below in relative order of concern. Once these concerns are addressed, this work has the potential to be a valuable advance in the field.

- Overall, the manuscript is very difficult to follow, with many key gaps in explaining the premise regarding the nature of the cellular and micromechanical aspects of the model system selected. A focus is given to primary lung fibroblasts, with brief mention of bronchial epithelial cells, but without explanation or a rationale provided as to the strategy for selecting healthy versus diseased cells, fibroblasts alone versus co-culture with epithelial cells, bronchial vs. Type I or Type II alveolar cells, potential introduction of a vascular / endothelial component. The same concern exists for the micromechanical structure – both from a materials and a mechanical / dimensional standpoint. These elements are largely introduced without explanation or rationale.
- In many instances, attempts to extrapolate these basic phenomena (tissue compaction/contractility) into larger scale properties of the lung (total lung capacity, honeycombing) seem to be premature, as the model is too early and too crude to try and force it to explain behavior that would likely require invoking more complexity.
- Pursuant to the above concern, there is a lack of forward-looking vision provided as to how this model would evolve into a more complex and powerful approach for mimicking key aspects of fibrotic diseases, and what the next steps are in the development of these systems.
- Many statements are provided without a sufficient basis, such as the claim that the design of the micropillar array prevents x-y contractile behavior while permitting such mechanisms in the z-direction. This will depend critically on the mechanical properties of the substrate from a design and modulus standpoint.
- Use of highly sorptive substrates such as PDMS appears inappropriate given the hydrophobic nature of the compounds being studied. As reported by Domansky (Microfluids Nanofluids 2017) PDMS adsorbs Pirfenidone quite strongly and therefore poor control over dosing would be expected.
- There is a general lack of schematics to show mechanistically how the platform technologies serve to evaluate the parameters of interest. Figure 3B does not do an adequate job of conveying the technical approach.
- In general, the manuscript comes across as a litany of results without a trajectory that focuses the reader on key conclusions. A narrower focus with more convincing results and explanation on

key observations regarding contractility, for instance, with greater attention to ensuring that the biological and mechanical models are well-justified and robust, would be more valuable.

Reviewer #2 (Remarks to the Author):

The manuscript "Fibrotic Microtissue Array to Predict Anti-Fibrosis Drug Efficacy" submitted by Asmani et al. describes the fabrication of lung microtissue comprised of lung fibroblasts within a collagen matrix, suspended over a micropillar device capable of measuring microtissue mechanical properties. The authors demonstrated that TGF- β treatment resulted in tissue stiffening and fibroblast activation and that anti-fibrosis treatments Pirfenidone and Nintedanib prevented these effects. The authors have formerly described the micropillar system and functional assays in previous manuscripts, so while the application of these microtissues for lung fibrosis modeling is novel, this is not a major breakthrough for the field. The ability to fabricate microtissues capable of measuring tissue stiffness as a predictive measure of fibrosis is advantageous. However this model still lacks many components that would provide greater similarity to in vivo lung testing. Additionally, there are serious concerns about the ability to extrapolate the small subset of mechanical measurements with in vivo clinical parameters such as total lung capacity.

Major comments:

1. The microtissue model is quite simplistic, comprising only of rat tail collagen type 1 and normal human lung fibroblasts. The authors do mention use of epithelial cells, however this was not used in combination with fibroblasts, nor for any functional testing, so it is unclear the relevance of this. Thus, the model lacks cell-cell interactions that are important for any pathological condition and could affect the fibrotic response and treatment. Further, lung connective tissue is composed of several different ECM components, so how was the decision made to create the microtissue only with collagen? Could decellularized tissue be utilized to obtain important cell-ECM interactions? Also, a microtissue without collagen, such as one with just elastin, would be another interesting test for comparison. This type of microtissue would also allow staining of all extracellular collagen for collagen deposition in comparison to cytosolic pro-collagen.
2. It is mentioned that the untreated and treated microtissues match the stiffness of healthy and fibrotic lung tissues, but the references utilized differ significantly along with the fact that one of the references is using murine lung tissue. The main point seems to be that there is a significant increase in stiffness from the untreated to treated, which is seen in fibrotic lung tissue.
3. While the derivations to calculate TLC may be mathematically sound based on the references, they could not be an accurate representation of TLC. Attempting to estimate total lung capacity of a microtissue without an actual capacity of air does not seem plausible. There are several factors that are in play for TLC, and a measurement of planar stiffness would not be able to convey all those factors when calculating TLC. It could be mentioned that based on the references that there is a relationship between increased stiffness and decreased TLC, but assigning total lung capacity values to a microtissues that do not have any capacity for air is not reasonable.
4. Even though fluorescence measurements were utilized for pro-collagen I, why were other ECM components not measured? It has been shown that fibrosis is associated with various protein deposition such as collagen I/III and fibronectin. The authors could conduct post-treatment ECM assays for ECM concentration comparisons between fibrotic and healthy microtissues to show increased deposition.
5. While it was interesting to see the reduction in dilation of the holes for the honeycombing model presented, attempting to relate this model to in vivo honeycombing is not reasonable as there are many other factors present in the process such as inflammation. This observation may be a useful parameter for the measurement of tissue stiffness and stress, however other than claiming that the hole dilation was induced and related to stress concentrations of the micropillars, extrapolation to clinical pathologies is not reasonable. The paper mentions a ring of myofibroblasts around each hole, but with the given immunofluorescence stain, it appears they are activated throughout most

of the tissue regardless.

6. While the authors claim that their system is capable of high-throughput drug analysis, it appears that only the microtissue fabrication approach could be utilized in a high-throughput manner. Much of the functional and phenotypic analysis seems to require manual imaging and off-board image analysis, which would significantly reduce throughput.

7. It is not clear why three and four leaflet designs were made, as square or rectangular designs were utilized for most of the data in this study. What is the relationship between the number of pillars, the angles between pillars and the output measurements. It is mentioned that the micropillars were strategically placed, how was this done (mathematically, computationally, etc.)? What exactly was the strategy?

Minor comments:

1. On line 127, it is mentioned that the fibroblasts are entangled throughout the microtissue thickness, but only reference an SEM image (Figure 1C). It would be better suited to reference Figure 1D additionally to show they are found throughout the thickness of the tissue. Still, the resolution in Figure 1D is too low to demonstrate nuclei throughout the thickness of the tissue.

2. Was there any additional preconditioning of the microtissues, prior to tensile testing besides stretching the length to 120% to account for hysteresis and relaxation? It is mentioned on line 449 that the microtissue device is capable of cyclic stretching, but was this utilized at all for the experiment or results?

3. For the honeycombing experiment, the contractile force was mentioned to be controlled by TGF- β treatment. What was the concentration of TGF- β used? Were varying concentrations tested to optimize contractile force?

4. It would be more helpful to provide equations and a more thoroughly explained process of how tensile force and stress were calculated in the methods section.

Reviewer #3 (Remarks to the Author):

In this manuscript, Asmani et al. engineered human lung microtissues for studying TGF β -induced fibrosis and the effect of two anti-fibrosis drugs. The manuscript is well written and the reported results are interesting to other researchers working in the field of biomedical engineering, lung biology and pathological tissue models. The present work is novel and well-fitted for Nature Communication, although some minor flaws need to be addressed before publication.

1. The statistics used in the manuscript need to be clarified and detailed. The authors state in the Materials and Methods section that "Data are presented as the mean with standard error bars showing the standard deviation" but explain in the figure caption that "Data are reported as the mean \pm SEM." Also the number of experiments and samples per experiment is required for assessing statistical differences between conditions.

2. The authors should indicate how the introduction of TGF β influence cell proliferation, tissue formation and organization. Similarly, the impact of Pirfenidone and Nintedanib on cell viability needs to be quantified in order to draw clear conclusions about their effect on tissue contractility.

3. The tissue-generated tension is quantified by measuring the micropillar deflection and using linear bending theory, which is only valid for small deformations. However, TGF β -treated samples already apply high forces, strongly bending the micropillars, before the authors stretched them for assessing the microtissue stiffness. It is thus unclear if the TGF β -treated samples are indeed stiffer or if it is only an artefact due to already strongly bent micropillars, out of the small deformation range.

February 28, 2018

RE: Decision on manuscript # NCOMMS-17-24025, entitled "Fibrotic microtissue array to predict anti-fibrosis drug efficacy"

Below we provide responses to the reviewers' comments and outline the changes we have made to the manuscript. These changes have been highlighted in both the main manuscript and supplemental material. We reproduce reviewers' comments in italics, followed by our responses.

Reviewer 1

• Overall, the manuscript is very difficult to follow, with many key gaps in explaining the premise regarding the nature of the cellular and micromechanical aspects of the model system selected. A focus is given to primary lung fibroblasts, with brief mention of bronchial epithelial cells, but without explanation or a rationale provided as to the strategy for selecting healthy versus diseased cells, fibroblasts alone versus co-culture with epithelial cells, bronchial vs. Type I or Type II alveolar cells, potential introduction of a vascular / endothelial component. The same concern exists for the micromechanical structure – both from a materials and a mechanical / dimensional standpoint. These elements are largely introduced without explanation or rationale.

We have performed new experiments and computational analyses to help explain the rationale for model selection. For cell type selection, we performed new tests on human lung small airway epithelial cells (SAECs, ATCC), the only commercially-available, non-cancerous, expandable lung epithelial cell that is anatomically close to the alveolar tissue, to study their effect on microtissue formation and fibrosis induction. Our data showed that while SAECs allowed microtissue formation, they did not respond to TGF- β 1 induced fibrotic transition (Supplemental material Fig. S6, 7). Lung bronchial epithelial cells such as BEAS-2B cells are anatomically far away and morphologically-different from the alveolar cells, and our tests showed that they also do not respond to TGF- β 1 stimulation. Commonly used lung alveolar cell lines such as A549 and NCI-H441 are cancerous cells, and commercially-available primary type I or type II alveolar cells are expensive and can not be expanded. Together, these issues with above lung epithelial cells make them unsuitable for use as models for anti-fibrosis drug screening. Since it is broadly accepted that regardless of the cell origin, the progression of lung fibrosis is predominantly contributed by differentiated myofibroblasts (Rock. PNAS 108, 2011; Liu. J Cell Biol 190, 2010), we chose to induce the myofibroblastic differentiation of lung fibroblast-populated microtissue and use it as a robust and effective

model for the study of anti-fibrosis drugs. We have included the above new results and rationale in the Results section and supplemental material.

Compared to currently used healthy fibroblasts, which can be induced into fibrosis in a controlled manner, diseased lung cells such as lung fibroblasts isolated from IPF patients suffer from donor variability and lack of control in the disease stage, and thus they are not suitable for use in the initial validation of the drug screening system. However, given their disease relevance, it would be interesting to include them in the microtissue system and examine their response to the anti-fibrosis drugs in the future, as an extended validation of the system. Although pericytes have been shown to be a potential source of fibrosis in several organs, their involvement in pulmonary fibrosis is not yet confirmed (Humphreys. Am J Pathol. 2010). Furthermore, in confirmed cases, pericytes contributed to fibrosis through differentiation to myofibroblasts, which current microtissue system has already correctly modeled. Therefore, to avoid unnecessary complexity, we did not include them in the current study. We have summarized these points as future work and included them in the Discussion section.

For the selection of microtissue design/geometry, we have developed finite element (FE) models and used these models to study the effects of microtissue design and boundary condition on the formation of biomimetic and structurally stable microtissues. In Figure 1 and corresponding Results section, we added new computational results showing that although large-size and medium-size designs both result in relatively high span length (S) to thickness (t) ratios (S/t ratio), which are ideal to model alveolar tissues, they are associated with different levels of stress concentrations that may affect the structural integrity of the microtissues differently. Experimental results further showed that the rate of microtissue failure increased with increased design complexity and microtissue size; therefore, we chose the medium-size, single-leaflet design for our drug screening system to ensure robust performance and easy handling. We also performed FE analyses to study the effects of microtissue design and boundary condition on the modeling of the biomechanics of traction force-induced bronchial dilation. In Figure 4 and corresponding Results section, we added new computational results showing that substantially increased micropillar rigidity can significantly increase the level of stress concentration, leading to the dilation of openings in the microtissue. This prediction was again validated experimentally. Together, these combined computational and experimental approaches allowed us to show the rationale for microtissue design selection.

• In many instances, attempts to extrapolate these basic phenomena (tissue compaction/contractility) into larger scale properties of the lung (total lung capacity, honeycombing) seem to be premature, as the model is too early and too crude to try and force it to explain behavior that would likely require invoking more complexity.

We thank the reviewer for pointing out the limitations in the original submission. We agree that estimation of the total lung capacity (TLC), despite being based on a published method, may not reflect the full complexity of TLC. We have replaced TLC measurement with tissue compliance measurement, which is a commonly measured tissue mechanical property for fibrosis evaluation and does not require mathematical extrapolation. We have added tissue compliance measurement in Fig. 3, 5 and 6 and in corresponding Results sections. We also agree that honeycombing is a complex pathological process and our presented opening formation in the fibrotic microtissue only models the mechanical aspects of

this process. Therefore, we have toned down our claim in Figure 4 and corresponding Results sections to “Modeling the biomechanics of traction bronchiectasis”.

- *Pursuant to the above concern, there is a lack of forward-looking vision provided as to how this model would evolve into a more complex and powerful approach for mimicking key aspects of fibrotic diseases, and what the next steps are in the development of these systems.*

We thank the reviewer for the constructive comments. In the future, we envision that diseased cells harvested from patients and vascular cells such as pericytes could be added to the microtissue culture to study the effects of these cells on fibrosis progression and treatment. Since the current mechanical stretching system is fully motorized, long-term, cyclic stretching can be applied to the microtissue culture to study the effect of physiologically-relevant mechanical stimuli on fibrosis progression and treatment. Furthermore, the utility validation of the current system will prepare it well for use as a screening system for testing the efficacy of potential anti-fibrosis compounds. Together, these future works have been proposed and discussed in the Discussion section.

- *Many statements are provided without a sufficient basis, such as the claim that the design of the micropillar array prevents x-y contractile behavior while permitting such mechanisms in the z-direction. This will depend critically on the mechanical properties of the substrate from a design and modulus standpoint.*

During microtissue seeding, the PDMS micropillars and microwell were rendered non-adhesive through the treatment of Pluronic F-127. This allowed the microtissue to compact freely in the z-direction under cell generated contraction. In the X-Y plane, the contraction was restricted by the micropillars due to their bending rigidity. To clarify this point and help the reader to understand the microtissue formation process, we have added two movies showing computationally simulated microtissue formation in 3D under cell-generated contraction and an experimental movie showing time-lapsed microtissue formation process in the Supplemental Material.

- *Use of highly sorptive substrates such as PDMS appears inappropriate given the hydrophobic nature of the compounds being studied. As reported by Domansky (Microfluids Nanofluids 2017) PDMS adsorbs Pirfenidone quite strongly and therefore poor control over dosing would be expected.*

We thank the reviewer for pointing out this potential problem. In the above paper (Domansky, Microfluids Nanofluids 2017), PBS solution containing drugs was made in contact with the PDMS specimen at 1: 1 volume ratio for 3-days, and it was found that 33% of Pirfenidone was recovered from the PBS solution. In our microtissue system, the diameter of the PDMS device chamber, which contains the microtissue array and holds drug-containing culture media, is 2.5 cm. According to the above paper, the penetration depth of drugs into the PDMS is about 0.5 mm after 3 days of absorption. Hence, the PDMS volume that absorbs drugs is around 250 mm³ in our system. However, the amount of culture media being added to the PDMS chamber is normally 2.5 mL = 2500 mm³, which is 10 times the volume of the absorptive PDMS. The amount of Pirfenidone being absorbed by the PDMS should be (1-33%) x 10% = 6.6% of the total amount over a 3 days period. Since we refresh culture media every two days, it is expected that the actual Pirfenidone being absorbed by the PDMS should be less than 5%, which is negligible.

- *There is a general lack of schematics to show mechanistically how the platform technologies serve to evaluate the parameters of interest. Figure 3B does not do an adequate job of conveying the technical approach.*

We thank the reviewer for pointing out this issue in the original submission. To help readers to understand better the working mechanism and work flow of our microtissue system, we have prepared new schematic drawings, flow charts and simulated plots of stress distribution and added them in all the figures. We also added descriptions to these schematics in the Results section to help with the understanding of the system.

- *In general, the manuscript comes across as a litany of results without a trajectory that focuses the reader on key conclusions. A narrower focus with more convincing results and explanation on key observations regarding contractility, for instance, with greater attention to ensuring that the biological and mechanical models are well-justified and robust, would be more valuable.*

We thank the reviewer for the constructive comments. With newly added schematics and flow charts, we hope the readers will understand better the flow of the manuscript and the connection between different parts of the results. We also hope that newly added FE simulation results, experimental results on lung epithelial cells and different ECM formulas and live-cell video on microtissue formation will help to convince the reviewer that the model selection process is well-justified and robust.

Reviewer 2

The manuscript “Fibrotic Microtissue Array to Predict Anti-Fibrosis Drug Efficacy” submitted by Asmani et al. describes the fabrication of lung microtissue comprised of lung fibroblasts within a collagen matrix, suspended over a micropillar device capable of measuring microtissue mechanical properties. The authors demonstrated that TGF- β treatment resulted in tissue stiffening and fibroblast activation and that anti-fibrosis treatments Pirfenidone and Nintedanib prevented these effects. The authors have formerly described the micropillar system and functional assays in previous manuscripts, so while the application of these microtissues for lung fibrosis modeling is novel, this is not a major breakthrough for the field. The ability to fabricate microtissues capable of measuring tissue stiffness as a predictive measure of fibrosis is advantageous. However this model still lacks many components that would provide greater similarity to in vivo lung testing.

We thank the reviewer for the critical comments. We understand that the reviewer had concerns over the novelty of the presented work due to the existence of earlier work that used similar micropillar system. However, we would like to point out that previously developed microtissues were thick and rod-shaped, and thus they were not able to recapitulate the membranous morphology of the lung alveolar tissue, which is critical for the development of unique pathological features in the fibrotic lung, such as the decline in tissue compliance and traction force-induced bronchial dilation. The suspended lung microtissue presented in the current study is the first engineered tissue model to recapitulate the unique membranous morphology of the human alveolar tissue; therefore, it represents a substantial advancement over previously published micropillar/microtissue models. The successful engineering of the membranous microtissue also opens up the possibility to model the physiology and pathology of other membrane-like tissue types, such as the fibrosis in retina and intestinal fibrosis associated with Crohns

disease. Furthermore, since the Huh/Ingber lung-on-a-chip model, being one of the most influential models in the field of diseased tissue modeling, lacks the abilities to model ECM remodeling and measure tissue mechanical properties, the current work significantly expands upon the value of existing alveolar disease models for fibrosis.

Major comments:

1. The microtissue model is quite simplistic, comprising only of rat tail collagen type 1 and normal human lung fibroblasts. The authors do mention use of epithelial cells, however this was not used in combination with fibroblasts, nor for any functional testing, so it is unclear the relevance of this. Thus, the model lacks cell-cell interactions that are important for any pathological condition and could affect the fibrotic response and treatment. Further, lung connective tissue is composed of several different ECM components, so how was the decision made to create the microtissue only with collagen? Could decellularized tissue be utilized to obtain important cell-ECM interactions? Also, a microtissue without collagen, such as one with just elastin, would be another interesting test for comparison. This type of microtissue would also allow staining of all extracellular collagen for collagen deposition in comparison to cytosolic pro-collagen.

We thank the reviewer for the constructive comments regarding the model selection. In the response to the first question raised by reviewer 1, we have detailed our strategy for cell type selection. Briefly, we have validated that non-cancerous, expandable lung epithelial cells (small airway epithelial cells) do not respond to TGF- β 1 induced fibrotic transition (Supplemental material Fig. S6, 7). Since the progression of lung fibrosis is predominantly contributed by differentiated myofibroblasts regardless of the cell origin (Rock. PNAS 108, 2011; Liu. J Cell Biol 190, 2010), we chose to induce the myofibroblastic differentiation of lung fibroblast-populated microtissue as a robust and effective model for the study of anti-fibrosis drugs. We have included the above new results and rationale in the Results section and supplemental material.

For the selection of the ECM components, we have performed new experiments using three different ECM formulas and compared their effects on microtissue formation and fibrosis induction. Comparison was made between microtissues fabricated using collagen, mixed ECM of collagen and fibrin and mixed ECM of collagen and matrigel, We found that there was no significant difference in either the expression of the biomarkers or the contractile force between these ECM formulas with or without TGF- β 1 treatment (Supplemental Material Fig. S8, 9), suggesting that different ECM formulas do not affect microtissue formation or fibrosis induction. Together, these results showed that lung fibroblast-populated collagen microtissue is a compositionally-simple yet effective model for lung interstitial fibrosis.

Current research on decellularized lung scaffolds focused on recellularizing it using its original tissue architecture (Crabbé. PLOS ONE 2015). Further research needs to be done to decompose decellularized lung scaffolds into ECM components for them to be used in reconstituted engineered tissues. While such approach is interesting and may be valuable to lung fibrosis research, we believe it is out of the scope of the current work. We agree that it would be interesting to test elastin in the microtissues; however, elastin is not a commonly-used ECM component that can be sourced commercially. In fact, we had difficulty to find a commercial supplier for elastin. We believe including such a rare research material in a drug testing system will significantly limit its potential widespread applications. We hope that newly added comparison results between commonly-used ECM formulas will help the readers to understand the effects of ECM components on the system performance.

Regarding the staining of ECM components to show matrix deposition, we have performed new experiments to stain ED-A fibronectin, a fibrosis-specific matrix marker, in the microtissue before and after anti-fibrosis treatment. We will explain this more in the response to question #4 in below. For collagen

deposition, it is known that fibroblasts deposit only minimal amounts of secreted collagen I into the extracellular matrices in vitro, due to the tardy procollagen C-proteinase/BMP1 activity under aqueous culture condition (Chen. Fibrogenesis Tissue Repair. 2009). BMP-1 cleaves the C-propeptides of type I procollagen during the synthesis of extracellular matrix collagen fibrils, and its delayed activity results in most of the unprocessed procollagen ending up in the culture media. Therefore, even though collagen deposition is often detected in the histology of in vivo fibrosis samples, it is not a common immunofluorescent staining target in in vitro fibrosis studies.

2. It is mentioned that the untreated and treated microtissues match the stiffness of healthy and fibrotic lung tissues, but the references utilized differ significantly along with the fact that one of the references is using murine lung tissue. The main point seems to be that there is a significant increase in stiffness from the untreated to treated, which is seen in fibrotic lung tissue.

We are pleased with reviewer's comment in the last sentence which correctly summarized the purpose for stiffness comparison for microtissues. Due to limited report on lung tissue stiffness in the literature, we have included the references of both human (Booth, Am J Respir Crit Care Med 2012) and murine (Liu, J Cell Biol 2010) lung tissues in the original submission. Comparison on the stiffness values reported for healthy and fibrotic lung tissues actually showed that these values matched well between these two references, and these values also matched with the stiffness of our healthy and fibrotic microtissues. Given the broad impact of these works – both have been cited more than 230 times – we feel it is necessary to include them and compare to them.

3. While the derivations to calculate TLC may be mathematically sound based on the references, they could not be an accurate representation of TLC. Attempting to estimate total lung capacity of a microtissue without an actual capacity of air does not seem plausible. There are several factors that are in play for TLC, and a measurement of planar stiffness would not be able to convey all those factors when calculating TLC. It could be mentioned that based on the references that there is a relationship between increased stiffness and decreased TLC, but assigning total lung capacity values to a microtissues that do not have any capacity for air is not reasonable.

We thank the reviewer for pointing out this limitation in the original submission. We agree that estimation of the total lung capacity (TLC), despite being based on a published method, may not reflect the full complexity of TLC. We have replaced TLC measurement with tissue compliance measurement, which is a commonly measured tissue mechanical property for fibrosis evaluation and does not require mathematical extrapolation. We have added tissue compliance measurement in Fig. 3, 5 and 6 and in corresponding Results sections. We have included this explanation in the responses to Reviewer 1's comments as well.

4. Even though fluorescence measurements were utilized for pro-collagen I, why were other ECM components not measured? It has been shown that fibrosis is associated with various protein deposition such as collagen I/III and fibronectin. The authors could conduct post-treatment ECM assays for ECM concentration comparisons between fibrotic and healthy microtissues to show increased deposition.

We thank the reviewer for the constructive comments. We have performed new experiments to stain for extracellular ED-A fibronectin, a fibrosis-specific matrix marker, in untreated, fibrosis-induced and anti-

fibrosis treated microtissues. Our results showed that fibrosis induction using TGF- β 1 caused substantial increase in the deposition of highly-aligned ED-A fibronectin fibers in the lung microtissue, and anti-fibrosis treatments at effective concentrations strongly inhibited the deposition of the ED-A fibronectin. Together, these results supported the conclusions drawn based on the immuno-staining of other fibrosis biomarkers such as α -SMA and pro-collagen. We have included the new ED-A fibronectin results in Figure 2, 5 and 6 and the corresponding Results sections.

5. While it was interesting to see the reduction in dilation of the holes for the honeycombing model presented, attempting to relate this model to in vivo honeycombing is not reasonable as there are many other factors present in the process such as inflammation. This observation may be a useful parameter for the measurement of tissue stiffness and stress, however other than claiming that the hole dilation was induced and related to stress concentrations of the micropillars, extrapolation to clinical pathologies is not reasonable. The paper mentions a ring of myofibroblasts around each hole, but with the given immunofluorescence stain, it appears they are activated throughout most of the tissue regardless.

We thank the reviewer for the critical comments. We agree that honeycombing is a complex pathological process and our presented opening formation in the fibrotic microtissue only models the mechanical aspects of this process. Therefore, we have toned down our claim in Figure 4 and corresponding Results sections to “Modeling the biomechanics of traction bronchiectasis”. We have included this explanation in the responses to Reviewer 1’s comments as well. We are glad that the reviewer noticed the widespread distribution of myofibroblasts in the long microtissue, which suggests successful induction of fibrosis. All these myofibroblasts contributed to the high stress level in the microtissue, but the ones around the holes contributed further to the hole dilation. We have added finite element simulation results showing that the distribution of the myofibroblasts around the holes matched with the distribution of the principal stress vectors in this area, proving the critical contribution to hole formation by the myofibroblasts. We have added these new simulation results in the Figure 4.

6. While the authors claim that their system is capable of high-throughput drug analysis, it appears that only the microtissue fabrication approach could be utilized in a high-throughput manner. Much of the functional and phenotypic analysis seems to require manual imaging and off-board image analysis, which would significantly reduce throughput.

We thank the reviewer for the critical comments. Our current system – 12 well plate with each well containing an array of 8 x 8 microtissues – is actually compatible with motorized microscopy which allows automated image acquisition and analysis. However, we agree that due to increased engineering complexity, our system may not offer the same level of throughput as 2D cultures in 96 well or 384 well plates. Therefore, we have toned down our statement and removed the high-throughput term from the manuscript.

7. It is not clear why three and four leaflet designs were made, as square or rectangular designs were utilized for most of the data in this study. What is the relationship between the number of pillars, the angles between pillars and the output measurements. It is mentioned that the micropillars were strategically placed, how was this done (mathematically, computationally, etc.)? What exactly was the strategy?

For the selection of microtissue design/geometry, we have developed finite element (FE) models and used these models to study the effects of microtissue design and boundary condition on the formation of biomimetic and structurally stable microtissues. In Figure 1 and corresponding Results section, we added new computational results showing that although large-size and medium-size designs both result in relatively high span length (S) to thickness (t) ratios (S/t ratio), which are ideal to model alveolar tissues, they are associated with different levels of stress concentrations that may affect the structural integrity of the microtissues differently. Experimental results further showed that the rate of microtissue failure increased with increased design complexity and microtissue size; therefore, we chose the medium-size, single-leaflet design for our drug screening system to ensure robust performance and easy handling. We also performed FE analyses to study the effects of microtissue design and boundary condition on the modeling of the biomechanics of traction force-induced bronchial dilation. In Figure 4 and corresponding Results section, we added new computational results showing that substantially increased micropillar rigidity can significantly increase the level of stress concentration, leading to the dilation of openings in the microtissue. This prediction was again validated experimentally. Together, these combined computational and experimental approaches allowed us to show the rationale for microtissue design selection. We have included this explanation in the response to Reviewer 1's comments as well.

Minor comments:

1. On line 127, it is mentioned that the fibroblasts are entangled throughout the microtissue thickness, but only reference an SEM image (Figure 1C). It would be better suited to reference Figure 1D additionally to show they are found throughout the thickness of the tissue. Still, the resolution in Figure 1D is too low to demonstrate nuclei throughout the thickness of the tissue.

We thank the reviewer for the constructive comments. We have updated the cross-sectional view (now Fig. 1G) of fibroblast-populated microtissue showing F-actin staining throughout the tissue thickness. Fig 1G is presented with high resolution to allow clear view of the cell/ECM distribution.

2. Was there any additional preconditioning of the microtissues, prior to tensile testing besides stretching the length to 120% to account for hysteresis and relaxation? It is mentioned on line 449 that the microtissue device is capable of cyclic stretching, but was this utilized at all for the experiment or results?

A recent publication on microtissue stretching/unstretching behavior from author's previous group showed that microtissue returned to its initial stress-strain state after a stretching/unstretching cycle and there was no residue plastic deformation (Liu. Scientific Reports 2016). This is because that the viscoplastic property of the ECM, which potentially contributes to the unrepeatable behavior during preconditioning of the native soft tissues, was shielded by the active dynamics of the cells in reconstituted microtissues. Given this finding, we did not perform preconditioning on our microtissues. We did not perform cyclic stretching in the current study; however, this is has been proposed as a future work in the Discussion section.

3. For the honeycombing experiment, the contractile force was mentioned to be controlled by TGF- β treatment. What was the concentration of TGF- β used? Were varying concentrations tested to optimize contractile force?

We used 5ng/mL TGF- β 1 in the tissue opening formation experiment, which is the same concentration as other experiments in the current study. This concentration is the most often used concentration found in the in vitro study of fibrosis. We have used this concentration to keep consistent with the literature.

4. It would be more helpful to provide equations and a more thoroughly explained process of how tensile force and stress were calculated in the methods section.

We thank the reviewer for the suggestion. Equation and diagram for microtissue contractile force calculation have been included in the supplemental material Fig. S5, and equation and diagram for microtissue stiffness calculation have been included in the new Figure 3. The method section was also updated to reflect these changes.

Reviewer 3

1. The statistics used in the manuscript need to be clarified and detailed. The authors state in the Materials and Methods section that “Data are presented as the mean with standard error bars showing the standard deviation” but explain in the figure caption that “Data are reported as the mean \pm SEM.” Also the number of experiments and samples per experiment is required for assessing statistical differences between conditions.

We thank the reviewer for pointing out this issue. We have updated our statement to “Data are presented as the mean with error bars showing the standard deviation” in the Methods section and changed the statement to “Data are reported as the mean \pm SD” in the figure captions. These statements reflect the true statistics reported in the results. We also updated the number of experiments and samples in the figure captions.

2. The authors should indicate how the introduction of TGF β influence cell proliferation, tissue formation and organization. Similarly, the impact of Pirfenidone and Nintedanib on cell viability needs to be quantified in order to draw clear conclusions about their effect on tissue contractility.

The effect of TGF- β 1 on cell proliferation has been well documented in existing literature, which showed that TGF- β 1 stimulates cell proliferation in 2D but not so much in 3D due to the spatial confinement to the cells by the ECM meshwork (Chen. Tissue Eng Part A. 2012, 18 (23-24)). In the current study, we tested the proliferation in the microtissue and we showed that cell proliferation under TGF- β 1 treatment is equivalent to that under untreated condition, but it was modestly inhibited by anti-fibrosis drugs. We have included these new results in supplemental material Figure S16.

The concentrations of Pirfenidone and Nintedanib used in the current study are commonly used concentrations in the literature that showed no toxicity to the cells (Wollin. Eur Respir J 2015). We have confirmed the impact of Pirfenidone and Nintedanib on cell viability by running Live/Dead assay under these drug treatments. We showed that the highest concentration of Pirfenidone and Nintedanib used in the current study did not cause significant cell death and the viability remained around 95%. We have included these results in the supplemental material Figure S14.

3. The tissue-generated tension is quantified by measuring the micropillar deflection and using linear bending theory, which is only valid for small deformations. However, TGF β -treated samples already apply high forces, strongly bending the micropillars, before the authors stretched them for assessing the microtissue stiffness. It is thus unclear if the TGF β -treated samples are indeed stiffer or if it is only an artefact due to already strongly bent micropillars, out of the small deformation range.

In our previous publication (Zhao, Biomaterials 2014), we have performed theoretical analyses to compare the micropillar load-deformation relationships derived using analytical small deflection theory, analytical large deflection theory and nonlinear FE analysis. We showed that the results calculated using small deflection theory agreed well with that calculated using large deflection theory when the deflection is less than 40% of the micropillar height. In our current system, the averaged micropillar deflection for TGF- β 1 treated sample is 75 μm and that for untreated sample is 35 μm . Under stretching, the additional micropillar deflection for TGF- β 1 treated sample is 15 μm and that for untreated sample is 5 μm . Together, the total micropillar deflection for TGF- β 1 treated sample under stretching is 90 μm , which is $90 \mu\text{m} / 270 \mu\text{m} = 33\%$ of the micropillar height (270 μm). Therefore, the force calculation using small deflection theory is a reasonable estimation.

With the above changes, we feel that reviewers' comments have been fully addressed and the manuscript has been substantially improved. We look forward to your comment and decision regarding the revision.

REVIEWERS' COMMENTS:

Reviewer #1 (Remarks to the Author):

The revised manuscript "Fibrotic microtissue array to predict anti-fibrosis drug efficacy" resolves most of the major concerns raised in the initial review, and the rebuttal letter provides a clear and comprehensive description of how each concern has been addressed, and a rationale for the various choices made in designing and executing the study. Only a few comments relative to the revised manuscript:

- 1) The most important aspect of the revision is the replacement of the modeling of total lung capacity and honeycombing with tissue compliance measurements and biomechanical aspects of bronchiectasis. This revised approach provides a bridge between the microscale model and phenomena that do not rely on features absent from the system as presented or dependent on larger scale mechanisms.
- 2) The additional experiments performed by the author team are significant and are very much appreciated. These experiments shed light on questions regarding the role of various cell types regarding response to TGF-B1 stimulation and other aspects of fibrosis induction. While these data are summarized in the Results and Discussion section, the authors are encouraged to briefly reference the rationale for the choice of cells in the Introduction.
- 3) Data on the limited effect of pirfenidone is helpful in alleviating the concern, but it is only shared in the rebuttal and not referenced in the revised manuscript. Other readers will be concerned about this potential issue.
- 4) The supplemental videos are of limited value, perhaps because they are not accompanied by sufficient documentation to explain what is being displayed and its meaning and relevance.

Reviewer #2 (Remarks to the Author):

Overall, the authors addressed a majority of the significant issues associated with the first draft of the manuscript. The manuscript focuses on the methodology and verification to fabricate a microtissue that exhibits a fibrotic phenotype within a micropillar system that allows both imaging analysis and mechanical analysis of the tissue. While the manuscript focuses on a simplistically designed microtissues seeded with lung fibroblast, the system could be expanded in the future for more complex designs, greater cell diversity, and increased biological components for improved comparison to in vivo subjects. Still, there are some minor comments that could still be addressed prior to acceptance.

Minor Comments

- 1) While the authors sufficiently responded to the high throughput capability of their system with regards to the imaging analysis of the device, is the system still capable of the stated higher capacity with regards to the mechanical stretching of the device? It appears that each 8x8 array within a 12 well plate will be stretched independently? Is this true and can this be clarified in the manuscript?
- 2) Another previous comment mentioned was the lack of the use of complex geometries for the majority of the experimental results. While it is clear why the medium-sized design was chosen, it should be mentioned that this may be a limitation of the system, as more complex geometries may be required for analysis of tissues such as lung. Additionally, are there any possible improvements to the design for future directions that could allow complex geometries?
- 3) Another aspect the authors addressed was the lack of a variety of ECM components within the microtissue. While the addition of fibrin and Matrigel did add some variety, elastin specifically, would provide more significant data as shown from previous articles (Blaauboer, Matrix Biology 2014). There are less conventional sources of elastin, such as that of bovine neck ligament from Sigma, that could be used for more robust experimentation. This could be a future direction for a

mixed-ECM microtissue or just simply an elastin-only microtissue for comparison.

Reviewer #3 (Remarks to the Author):

In the revised manuscript by Asmani et al., the authors have answered the reviewers' questions. Although the presented tissue model is somehow simplistic in comparison with a native lung tissue, the ability of this bottom-up approach to quantitatively recapitulate the early events of fibrogenesis and demonstrate the impact of anti-fibrosis drugs in a lung fibroblast-populated microtissue justifies its publication in Nature Communication.

April 5, 2018

RE: Final revision for manuscript # NCOMMS-17-24025B, entitled "Fibrotic microtissue array to predict anti-fibrosis drug efficacy"

Below we provide responses to the reviewers' comments and outline the changes we have made to the manuscript. These changes have been tracked in both the main manuscript and supplementary files. We reproduce reviewers' comments in italics, followed by our responses.

Reviewer 1

• The revised manuscript "Fibrotic microtissue array to predict anti-fibrosis drug efficacy" resolves most of the major concerns raised in the initial review, and the rebuttal letter provides a clear and comprehensive description of how each concern has been addressed, and a rationale for the various choices made in designing and executing the study. Only a few comments relative to the revised manuscript:

1) The most important aspect of the revision is the replacement of the modeling of total lung capacity and honeycombing with tissue compliance measurements and biomechanical aspects of bronchiectasis. This revised approach provides a bridge between the microscale model and phenomena that do not rely on features absent from the system as presented or dependent on larger scale mechanisms.

We are pleased by reviewer's positive comments regarding the two important improvements made in the revision. As appreciated by the reviewer, these improvements enabled connection between microscale model and basic physiopathological phenomena of lung fibrosis.

2) The additional experiments performed by the author team are significant and are very much appreciated. These experiments shed light on questions regarding the role of various cell types regarding response to TGF-B1 stimulation and other aspects of fibrosis induction. While these data are summarized in the Results and Discussion section, the authors are encouraged to briefly reference the rationale for the choice of cells in the Introduction.

We appreciate reviewer's positive comments on the additional experiments. We have now included a brief new section in the Introduction to explain the rationale for cell type selection.

3) Data on the limited effect of pirfenidone is helpful in alleviating the concern, but it is only shared in the rebuttal and not referenced in the revised manuscript. Other readers will be concerned about this potential issue.

We thank reviewer's positive comments on our analysis that showed limited pirfenidone absorption in our PDMS device. We have now included this analysis as a Supplementary Note.

4) The supplemental videos are of limited value, perhaps because they are not accompanied by sufficient documentation to explain what is being displayed and its meaning and relevance.

We thank reviewer's comments regarding the supplementary movies. We have now added more detailed explanations to the movies and hope they will help the audience to understand the movies better.

Reviewer 2

Overall, the authors addressed a majority of the significant issues associated with the first draft of the manuscript. The manuscript focuses on the methodology and verification to fabricate a microtissue that exhibits a fibrotic phenotype within a micropillar system that allows both imaging analysis and mechanical analysis of the tissue. While the manuscript focuses on a simplistically designed microtissues seeded with lung fibroblast, the system could be expanded in the future for more complex designs, greater cell diversity, and increased biological components for improved comparison to in vivo subjects. Still, there are some minor comments that could still be addressed prior to acceptance.

Minor Comments

1) While the authors sufficiently responded to the high throughput capability of their system with regards to the imaging analysis of the device, is the system still capable of the stated higher capacity with regards to the mechanical stretching of the device? It appears that each 8x8 array within a 12 well plate will be stretched independently? Is this true and can this be clarified in the manuscript?

We thank the reviewer for the positive comments regarding the high throughput capability of the system. For the stretching capacity of the device, each 8x8 microtissue array within a 12 well plate was stretched independently, which represents a significantly higher throughput than conventional tissue stretching experiments. We have included a short explanation in the Method section to clarify this point.

2) Another previous comment mentioned was the lack of the use of complex geometries for the majority of the experimental results. While it is clear why the medium-sized design was chosen, it should be mentioned that this may be a limitation of the system, as more complex geometries may be required for analysis of tissues such as lung. Additionally, are there any possible improvements to the design for future directions that could allow complex geometries?

We thank the reviewer for the constructive comments regarding the geometrical design of the microtissues and we have included a short statement in the Discussion section to indicate this limitation of the medium-sized design. Regarding design improvement, it would be possible to use wide microwalls in place of currently used micropillars to increase the contact area between the tissue and PDMS supports. This will help to reduce the stress concentration in the microtissue and thus increase the success rate for designs with complex geometries. We have included this point as a future direction in the Discussion section.

3) Another aspect the authors addressed was the lack of a variety of ECM components within the microtissue. While the addition of fibrin and Matrigel did add some variety, elastin specifically, would provide more significant data as shown from previous articles (Blaauboer, Matrix Biology 2014). There are less conventional sources of elastin, such as that of bovine neck ligament from Sigma, that could be used for more robust experimentation. This could be a future direction for a mixed-ECM microtissue or just simply an elastin-only microtissue for comparison.

We thank the reviewer for providing the reference and the source for elastin. We agree that using elastin in the current system will shed new light on the cell-ECM interaction involved in fibrosis and further expand physiological relevancy of the current system. We have now included this statement and the above reference as a future direction in the Discussion section.

Reviewer 3

In the revised manuscript by Asmani et al., the authors have answered the reviewers' questions. Although the presented tissue model is somehow simplistic in comparison with a native lung tissue, the ability of this bottom-up approach to quantitatively recapitulate the early events of fibrogenesis and demonstrate the impact of anti-fibrosis drugs in a lung fibroblast-populated microtissue justifies its publication in Nature Communication.

We thank the reviewer for the positive comments.

With the above changes, we feel that reviewers' comments have been fully addressed and the manuscript has again been improved. We look forward to your comment and decision regarding the revision.